# Structural basis of dimerization of chemokine receptors CCR5 and CXCR4

**Daniele Di Marino**[1,2,3,6]**, Paolo Conflitti** ●[4,6]**, Stefano Motta** ●[5,6] **&
Vittorio Limongelli** ●[4] ✉

G protein-coupled receptors (GPCRs) are prominent drug targets responsible for extracellular-to-intracellular signal transduction. GPCRs can form functional dimers that have been poorly characterized so far. Here, we show the dimerization mechanism of the chemokine receptors CCR5 and CXCR4 by means of an advanced free-energy technique named coarse-grained metadynamics. Our results reproduce binding events between the GPCRs occurring in the minute timescale, revealing a symmetric and an asymmetric dimeric structure for each of the three investigated systems, CCR5/CCR5, CXCR4/CXCR4, and CCR5/CXCR4. The transmembrane helices TM4-TM5 and TM6-TM7 are the preferred binding interfaces for CCR5 and CXCR4, respectively. The identified dimeric states differ in the access to the binding sites of the ligand and G protein, indicating that dimerization may represent a fine allosteric mechanism to regulate receptor activity. Our study offers structural basis for the design of ligands able to modulate the formation of CCR5 and CXCR4 dimers and in turn their activity, with therapeutic potential against HIV, cancer, and immune-inflammatory diseases.

G protein-coupled receptors (GPCRs) are cellular membrane proteins regulating the signal transduction from the extracellular to the intracellular environment[1]. They represent the largest and pharmacologically most relevant protein family, coded by ~4% of the protein-coding genome and targeted by ~34% of marketed drugs[2]. Despite their low sequence similarity, GPCRs share a common barrel tertiary structure consisting of seven transmembrane helices (TM1–TM7) that host the orthosteric binding site—the binding site of endogenous ligands—on the extracellular side and the binding site of the effector G protein on the intracellular side[3]. During the last three decades, growing evidence has demonstrated that GPCRs are able to work as dimers and oligomers, in addition to the single protomer form, paving the way to studies on the physiological and pathological role of receptors association[4–8].

Important members of the GPCR family are the chemokine receptors CCR5 and CXCR4[9], which regulate crucial processes like leukocyte migration and whose malfunctioning is associated with cancer, immune and neurodegenerative diseases[10,11]. Furthermore, these two receptors play a key role in the human immunodeficiency virus type 1 infection (HIV-1)[11–14], acting as the first recognition site for the virus on the host cell surface[15,16]. Targeting CCR5 and CXCR4 is an attractive strategy to block the virus entry into the host cell that led to the discovery of the allosteric CCR5 inhibitor maraviroc, approved for anti-HIV-1 treatment[17–20].

Despite the wealth of experimental data, including the atomistic structures of these two receptors[21–25], many aspects of their functional mechanism remain poorly understood. Among them, the most relevant is the ability of CCR5 and CXCR4 to form functional hetero- and

[1]Department of Life and Environmental Sciences - New York-Marche Structural Biology Centre (NY-MaSBiC), Polytechnic University of Marche, Via Brecce Bianche, 60131 Ancona, Italy. [2]Neuronal Death and Neuroprotection Unit, Department of Neuroscience, Mario Negri Institute for Pharmacological Research-IRCCS, Via Mario Negri 2, 20156 Milan, Italy. [3]National Biodiversity Future Center (NBFC), Palermo, Italy. [4]Università della Svizzera italiana (USI), Faculty of Biomedical Sciences, Euler Institute, Via G. Buffi 13, CH-6900 Lugano, Switzerland. [5]Department of Earth and Environmental Sciences, University of Milano-Bicocca, Piazza della Scienza 1, 20126 Milan, Italy. [6]These authors contributed equally: Daniele Di Marino, Paolo Conflitti, Stefano Motta. ✉e-mail: vittoriolimongelli@gmail.com

homodimers[26–29], which plays a specific role in regulating receptor activities like signal transduction, receptor trafficking, and internalization[30,31]. Spectroscopic and mutagenesis experiments have suggested potential models of CCR5 and CXCR4 dimers as well as for other GPCRs, like opioid and adrenergic receptors[16,27,29,32–35]. Particularly for CXCR4, recent works showed a correlation between the expression levels of the receptor with the dimer formation[36,37], whereas specific CXCR4 ligands−AMD3100, IT1t−can modulate the dimer vs. monomer equilibrium[36–38]. However, a comprehensive structural and functional characterization of receptor dimers is still lacking[39]. Therefore, the fascinating scenario of having dimers of CCR5 and CXCR4 as pharmacological targets is counteracted by the lack of knowledge on even basic aspects of the receptors interaction, like the dimerization interface, the stability, and the dynamics of the dimer complexes.

In this work, we tackle these challenges by investigating in silico the molecular binding mechanism of CCR5 and CXCR4 in the membrane environment. In detail, we generate a model in which the receptors are immersed in a POPC phospholipid bilayer with 10% cholesterol. In this environment, the receptors are fully free to diffuse in the membrane plane, and their binding interaction was investigated in the minute timescale by means of a multiscale free-energy technique named Coarse-Grained MetaDynamics (CG−MetaD)[40]. Such a method was originally employed by our group to elucidate the dimerization mechanism of the epidermal growth factor receptor[40]. Here, we build up three systems in which we investigate the formation of (i) CCR5 dimers; (ii) CXCR4 dimers; and (iii) CCR5−CXCR4 heterodimers. After a total of 5.5 ms of enhanced sampling simulations−corresponding to minutes in real time−we can thoroughly characterize the free-energy landscape of the receptor dimerization process, providing structural insight at a long time scale and high spatial resolution. This allows disclosing the receptor dimeric structures as the lowest energy−hence most probable−states and the way the receptors interact during the dimerization process, including energetic metastable states and the role played by phospholipids, water, and cholesterol molecules.

In all the investigated systems−CCR5 dimer, CXCR4 dimer, and CCR5−CXCR4 heterodimer−the receptors assume two dimeric structures, one with a symmetric and the other with an asymmetric binding mode. The transmembrane helices TM4−TM5 and TM6−TM7 are detected as the preferred binding interfaces for CCR5 and CXCR4, respectively. This finding is also confirmed in the active form of CCR5− i.e., in complex with the agonist Chemokine C−C Motif Ligand 3 (CCL3) and G protein−which binds to the other protomer, CCR5 or CXCR4, always through TM4−TM5, suggesting a dimerization interface selection mechanism mediated by the G protein. Our results further indicate that dimerization may represent a fine allosteric mechanism in which the activation of one protomer might be modulated by the interaction with a second protomer[6,41–46]. This is the case of specific homo- and heterodimeric structures where one protomer assumes a more active state or shifts towards the inactive form, thus hampering the G protein binding.

The identified dimeric structures of CCR5, CXCR4, and CXCR4−CCR5 are assessed in a realistic plasma membrane model, asymmetrically composed by differently saturated phospholipids and higher cholesterol concentration, thus mimicking the composition of in vivo cell membranes[47], and finally refined by atomistic Molecular Dynamics (MD) simulations. Our dimeric structures pave the way to the structure-based design of ligands capable of modulating the formation of CCR5 and CXCR4 dimers and, in turn, their activity, with therapeutic potential against HIV, cancer, and immune-inflammatory diseases related to these chemokine receptors.

## Results
### Binding free energy of CCR5 and CXCR4 dimers
The CCR5 and CXCR4 homodimerization and CCR5−CXCR4 heterodimerization were investigated in three different systems using CG−MetaD in which the studied process is accelerated by adding a bias potential on two selected system's degrees of freedom, named collective variables (CVs). These are: (i) the distance between protomers (CV1); and (ii) the torsion angle defining the orientation of one protomer relative to the other (CV2) (see "Methods" and Supplementary Fig. 1). In order to favor data transparency and reproducibility of our results, we made the input files of our simulations available in PLUMED-NEST, a public repository of simulation files we have recently published[48]. For the sake of clarity, hereafter, we label residues and helices belonging to the first protomer with superscript $a$ and those belonging to the second protomer with superscript $b$. In the case of the heterodimer, CCR5 is the first protomer, whereas CXCR4 is the second one.

The two protomers were embedded in a square membrane of 400 nm², composed of POPC and cholesterol molecules in a 9:1 ratio. The simulations started with the two protomers in the fully unbound state, placed at ~7 nm each other, where no inter-protomer contact occurs. The binding free-energy calculation reached convergence at different simulation times in the three systems: 1.4 ms in CCR5−CCR5, 2.5 ms in CXCR4−CXCR4, and 1.6 ms in CCR5−CXCR4, for a total of 5.5 ms (Supplementary Fig. 2). Considering the sampling acceleration of our method in the order of magnitude 5−6−given by the combination of coarse-grained molecular dynamics with metadynamics−we could reproduce binding events occurring in the minute timescale. This allowed observing hundreds of back-and-forth events between the bound (i.e., dimer) and the unbound (i.e., monomer) states of the protomers (Supplementary Table 1). In each system, at the end of the simulation, we computed the binding free-energy surface (BFES) (Fig. 1). For the sake of discussion, we indicate in all the BFESs three regions based on the distance CV: (i) the bound; (ii) the pre-bound; and (iii) the unbound state (Fig. 1).

At the bound state, we characterized two dimeric structures for each system (six in total) that represent the lowest free-energy states (Fig. 2, definition in Supplementary Table 2). The first and very interesting finding is that in all three systems, one of the two lowest energy minima shows a symmetric binding mode−i.e., the two protomers interact through the same helices−while the other one has an asymmetric binding mode (Fig. 2). The atomistic structures of the dimers were retrieved from the coarse-grained representation using a backmapping procedure[49] and assessed by means of 3 μs atomistic molecular dynamics calculations (see Supplementary Fig. 3 and Supplementary Movie 1). The final six dimeric structures (2 per system) are released as PDB files in the Supplementary Materials and at www.pdbdb.com.

At the unbound state, the BFES is flat, characterized by several position-independent isoenergetic states as expected when the two protomers are not in contact. The calculated absolute binding free energy is −22.2 kcal/mol for the CCR5 homodimer, −21.1 kcal/mol for the CXCR4 homodimer, and −24.5 kcal/mol for CCR5−CXCR4 heterodimer (see "Methods" and Supplementary Fig. 2A), which are in line with previous estimates reported in literature[50–53]. We note that our simulation model embeds 2 receptors in a membrane of 400 nm², resulting in an approximate density of ~5000 receptors per μm². Interestingly, Isbilir et al.[36] found that CXCR4 shows a highly dimeric tendency already at receptor densities higher than 70 receptors per μm², while the monomeric state is favored at receptor densities below 0.3 receptors per μm². Consistent with these findings, our results confirm that at high receptor density, these GPCRs prefer forming dimers rather than remaining in the monomeric state. A similar trend was also found for the same receptors in T cell experiments where dimeric and oligomeric forms were largely more present than the monomeric state[26,27]. However, we note that comparing data from different studies is not trivial since factors like receptor density (e.g., receptor expression in diverse tissues) and the membrane composition (lipid types and presence of diverse proteins) might influence the receptor's binding interaction. In

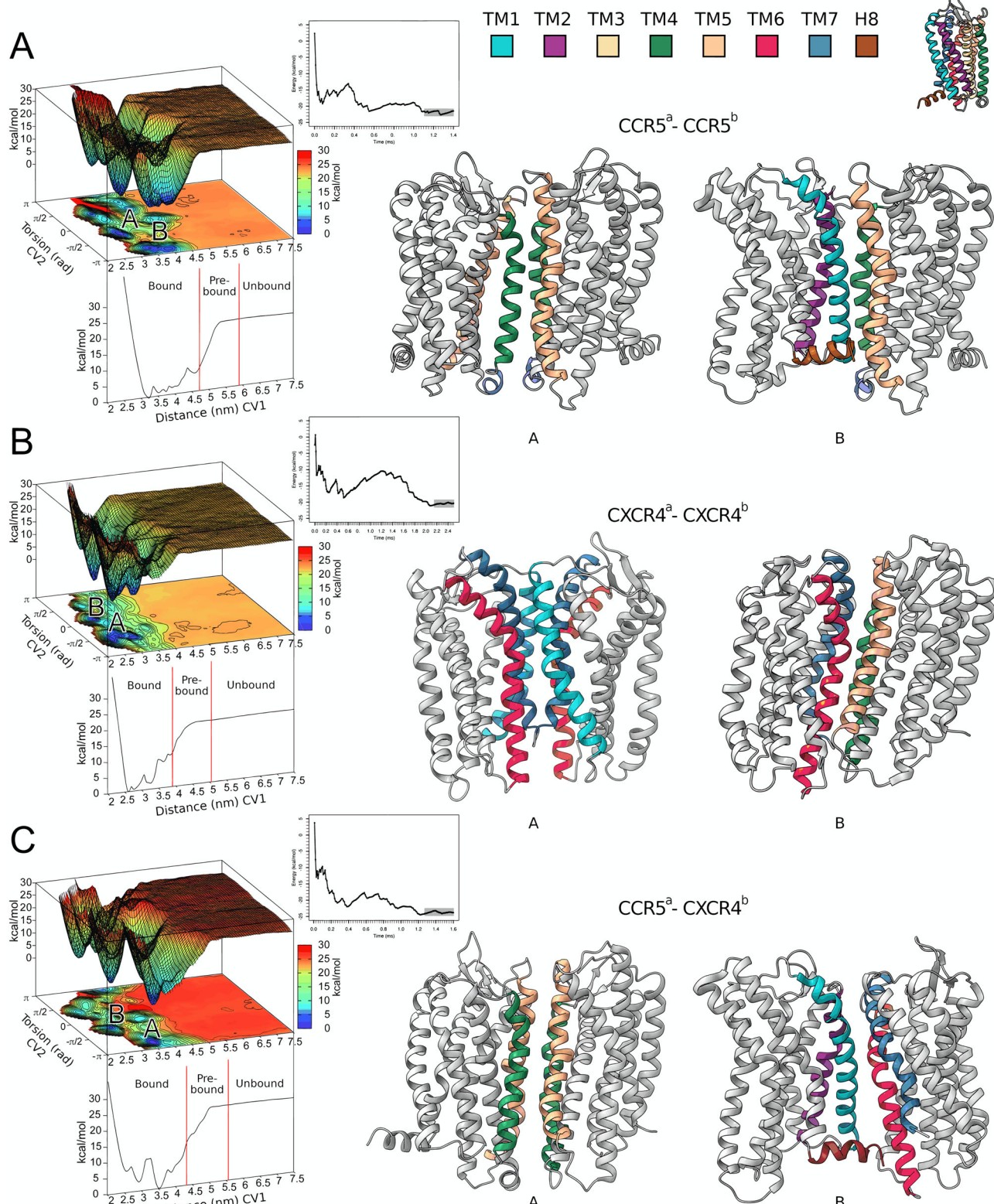

**Fig. 1 | Structures of CCR5 and CXCR4 homodimer and CCR5–CXCR4 heterodimer. A** Binding free energy surface (BFES) of CCR5 homodimerization. **B** BFES of CXCR4 homodimerization. **C** BFES of CCR5−CXCR4 heterodimerization. The convergence of the free-energy calculation between bound and unbound states is shown as inset for each system. The atomistic structures of the lowest energy dimeric states for each system are displayed on the right as gray cartoons with the transmembrane helices at the interface colored according to the color code reported at the top of the figure.

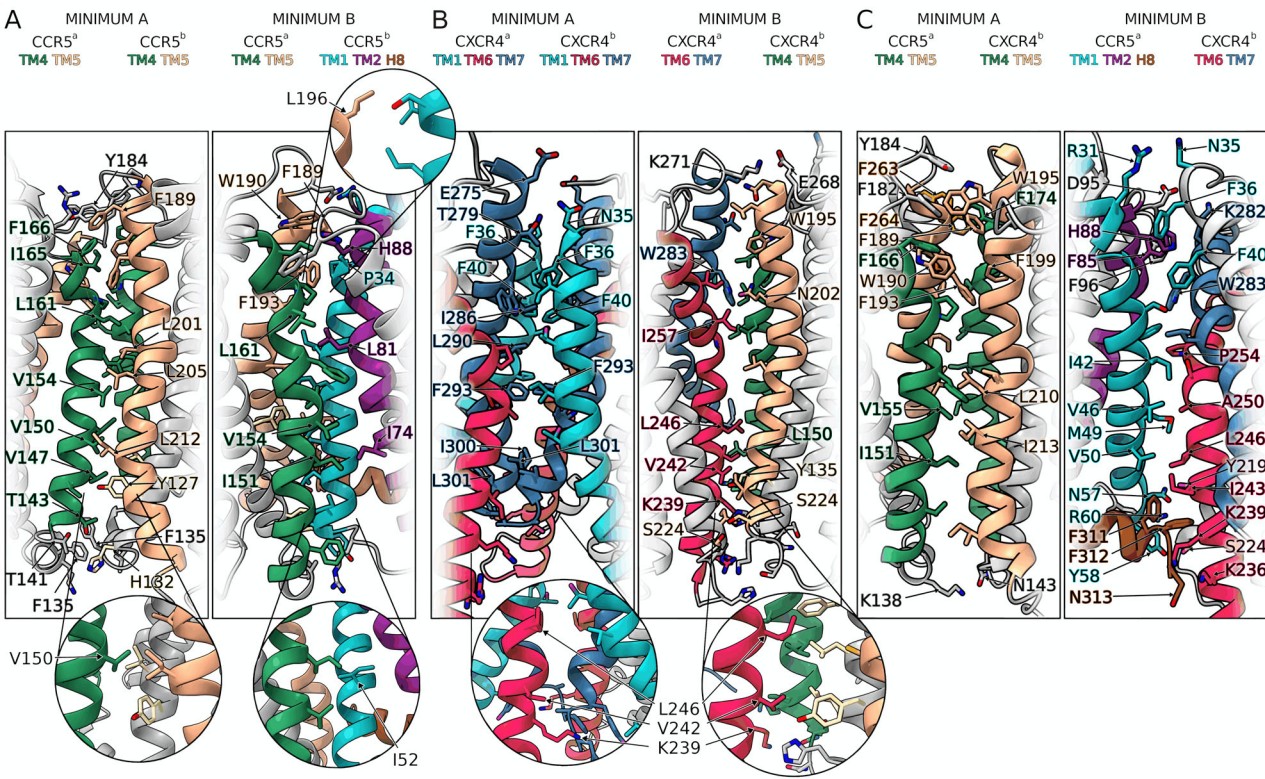

**Fig. 2 | Binding modes of CCR5 and CXCR4 homodimer and CCR5–CXCR4 heterodimer. A** CCR5 dimer binding interface. **B** CXCR4 dimer binding interfaces. **C** CCR5–CXCR4 heterodimer binding interfaces. Insets highlight residues identified by mutagenesis experiments that are involved in dimer formation. The TMs are colored according to the color code reported in Fig. 1.

addition, the accuracy of the experimental technique—in our case, the simulation model—should be also considered. Here, we employed the Martini coarse-grained force field[54–56], which is known to be enthalpically driven with entropy loss due to the coarse-grained representation of the system[57–59]. This leads to over-stabilization of the bound states. The employment of metadynamics in Martini coarse-grained simulations alleviates the issue, allowing the exploration of low probability regions[60–63]. As a result, the diffusion of the proteins in the membrane is enhanced and the correct interaction between the proteins is preserved, including the structural and energetic information for high-energy states and unbound states. For this reason, CG–MetaD has been successfully used by us and other research groups to study protein/protein interactions[40,64,65]. The use of CG–MetaD ensures recrossing events between receptor dimeric and monomeric states and a full exploration of the phase space (i.e., all the possible dimeric states), leading to the convergence of free-energy calculation and a quantitatively characterized free energy surface. The latter is necessary to disclose the low energy—hence most probable—dimeric states and the binding mechanism of these receptors, which is the aim of the present study. To this end, the relative free-energy difference between two or more dimeric states allows the identification of the lower energy dimers, whereas the absolute binding free energy value has minor significance since it can be influenced by simulation conditions, including the accuracy of the force field. A detailed description of each dimeric structure is reported in the next section.

## CCR5 homodimer

Two lowest energy receptor dimeric structures, A and B, are found in the BFES and reported in Fig. 1A and Fig. 2A. The first one shows a symmetric interaction mode involving TM4–TM5 (Fig. 2A, minimum A), while the second one reveals an asymmetric interaction mode involving TM4$^a$–TM5$^a$ and TM1$^b$–TM2$^b$–H8$^b$ (Fig. 2A, minimum B).

Dimeric structure A − In the lowest energy minimum A (see Source Data), the helices of the two protomers involved in the binding are the same (TM4–TM5). However, the orientation of the receptors at the dimerization interface is different. TM4$^a$ is parallel to TM5$^b$, whereas TM5$^a$ and TM4$^b$ are slightly tilted due to the rotation of ~10 degrees of protomer $a$ with respect to the membrane plane. For this reason, the TM5$^a$–TM4$^b$ contacts are lost toward the intracellular side (Fig. 2A, minimum A). This dimeric structure is stabilized by a hydrophobic cluster engaged by residues of the extracellular loops (ECL) 2 and the upper parts of TM4–TM5 of each protomer (Fig. 2A, minimum A). Among these residues are Ile165$^{a\ 4.62}$, Phe166$^{a\ 4.63}$, Phe182$^{a\ ECL2}$, Trp190$^{a\ 5.34}$ and Ile165$^{b\ 4.62}$, Phe166$^{b\ 4.63}$, Tyr184$^{b\ 5.28}$, Phe189$^{b\ ECL2}$ (Fig. 2A, minimum A and Supplementary Fig. 5A). At transmembrane level, a number of hydrophobic residues belonging to TM4 and TM5 form a zip-like interaction pattern that stabilizes the helix-helix interface. This is composed of a group of leucine and valine residues facing each other such as Val147$^{a\ 4.44}$, Val150$^{a\ 4.47}$, Val154$^{a\ 4.51}$, Leu161$^{a\ 4.58}$ on TM4$^a$ and Leu201$^{b\ 5.45}$, Leu205$^{b\ 5.49}$, Leu212$^{b\ 5.56}$ on TM5$^b$. At intracellular level, the intracellular loops (ICL) 2 of the two protomers interact by means of a hydrogen bond network formed by Thr141$^{a\ 4.38}$, Thr143$^{a\ 4.40}$ and Tyr127$^{b\ 3.51}$, His132$^{b\ 3.56}$, and a stable π-stacking engaged by Phe135$^{a\ ICL2}$ (Fig. 2A, minimum A and Supplementary Fig. 5A) and Phe135$^{b\ ICL2}$ (Supplementary Fig. 6).

Dimeric structure B − The second energy minimum B (see Source Data) represents a more compact dimeric structure with respect to A, characterized by an asymmetric dimerization interface (Fig. 2A, minimum B). On the extracellular side, Phe189$^{a\ 5.33}$, Trp190$^{a\ 5.34}$ and Phe193$^{a\ 5.37}$ on TM4$^a$–TM5$^a$ form a pocket of aromatic residues that hosts Pro34$^{b\ 1.36}$ and His88$^{b\ 2.62}$ of TM2$^b$. Furthermore, TM4$^a$ and TM1$^b$–TM2$^b$ assume an orthogonal orientation with respect to the membrane plane, engaging a number of inter-protomer hydrophobic interactions through residues like Ile151$^{a\ 4.48}$, Val154$^{a\ 4.51}$, and Leu161$^{a\ 4.59}$ with Ile74$^{b\ 2.48}$ and Leu81$^{b\ 2.55}$ (Fig. 2A, minimum B and Supplementary Fig. 5B).

The two identified dimeric structures of CXCR5 allow us to explain how specific residues—identified by mutagenesis experiments—affect the dimer formation. For instance, mutation of residues like Ile52[1.54], Val150[4.47], Leu196[5.40], Ile200[5.44], and Leu205[5.49] hampers the formation of CCR5 dimers, also playing a role in ligand binding to this receptor as in the case of CCL3[13,29,32]. The dimeric structures resolved by our calculations support and rationalize these data, showing such residues involved in interactions that stabilize both the dimer structures A and B (Fig. 2A). Indeed, introduction of mutations I52V[1.54], V150A[4.47], L196K[5.40], I200K[5.44], and L205K[5.49] (Supplementary Fig. 7A) on both minima resulted in a destabilizing effect in terms of protein-protein binding energy, with a $\Delta\Delta G$ value higher than 2.5 kcal/mol for minimum A and about 2–3 kcal/mol for minimum B (Supplementary Table 4). Furthermore, we computed the difference of the receptor Solvent Accessible Surface Area ($\Delta$SASA) between the monomeric and the dimeric states for both structures A and B. We found that dimer A has a lower $\Delta$SASA value if compared to B (A = 16.60 nm;² B = 18.40 nm²). These data confirm that the CCR5 protomers are more packed in B than in A. However, the lower free-energy value of A indicates that stronger interactions occur in dimer A. These include the previously described inter-protomer contacts but also interactions with neighboring environment molecules like phospholipids, cholesterol, and water. Such aspects are further discussed in the Supplementary discussion "The role of cholesterol in CCR5 and CXCR4 dimerization".

## CXCR4 homodimer

As in the case of CCR5, CXCR4 shows two lowest energy dimer structures A and B with a symmetric and asymmetric binding mode, respectively (Fig. 1B and Fig. 2B, minimum A and B). The two structures are energetically equivalent, having very close free-energy values (−21.1 and −20.8 kcal/mol for A and B, respectively). The binding interface in dimeric structure A is composed of TM1, TM6, and TM7, whereas dimer B comprises TM6[a]–TM7[a] and TM4[b]–TM5[b] (Fig. 2B).

Dimeric structure A − In dimer A (see Source Data), TM7 lies at the center of the dimeric structure in both protomers, sandwiched between TM1 and TM6. Here, a number of specular—symmetric— interactions between the protomers stabilizes the dimer (Fig. 2B and Supplementary Fig. 8A). Among these, the hydrophobic packing engaged by Ile286[7.37], Leu290[7.41], Phe293[7.44], Phe36[1.30], and Phe40[1.34] extends from the center to the extracellular portion of the protomers. At the extracellular level, Asn35[1.29] can form H-bonds with Glu275[7.26] and Thr279[7.30]. On the other end, at the intracellular level, the symmetry of the inter-protomer interaction is lost and the two receptors are slightly more distant. Here, it is interesting to note that Cys296[a 7.47] and Cys251[b 6.47] are in a position competent for formation of a disulfide bridge (Supplementary Fig. 9A). This finding suggests that employing redox techniques like cysteine cross-linking experiments might be suitable to further investigate such dimeric structures.

Dimeric structure B − In dimer B (See Source Data), TM6[a] protrudes between TM4[b]–TM5[b] where it engages a significant number of hydrophobic contacts (Fig. 2B and Supplementary Fig. 8B). An important contribution to the stability of this dimer comes from the salt bridge formed at ECL3 level between Glu268[b ECL3] and Lys271[a ECL3]. At the intracellular level, ICL3 is rich in polar residues that establish a network of H-bonds in addition to the specular H-bond engaged by Ser224[a 5.63] and Ser224[b 5.63]. Interestingly, in this dimeric structure, we found a cholesterol molecule placed in between the two protomers where it engages a series of hydrophobic interactions with residues like Leu132[a 3.48], Val214[a 5.53], Leu216[b 5.55], Leu246[b 6.42], and Phe249[b 6.45] (Fig. 3). The polar head of cholesterol forms an H-bond with Tyr135[a 3.51] that further stabilizes its binding mode (Fig. 3). This finding confirms that cholesterol might play an important role in mediating and stabilizing receptor dimerization as seen in other GPCRs[50,66]. Similarly to what was found in dimer A, we found that two cysteines, Cys 220[5.59] in

both protomers, are in a position competent for the formation of a disulfide bridge between the protomers (Supplementary Fig. 9B).

Previous studies on dimerization and oligomerization of CXCR4 showed the involvement of TM4 at the dimer interface[14,50,51,67]. Additional experiments indicated that TM4-derived peptides reduce—but not abolish—the ability of CXCR4 to form homodimers[14,67]. Taken together, these data suggest that TM4 participates in CXCR4 dimerization. However, alternative dimeric conformations not involving TM4 might also exist. This scenario is confirmed by our results that show TM4 is involved in one of the two possible dimeric structures (dimer B). In addition, dimer B sees the presence of residue Trp195[5.34] on TM5 at the binding interface (Fig. 2B), in agreement with previous data reporting the involvement of such amino acid in inter-protomer interactions[21,37]. Furthermore, residues of TM6 engage favorable interactions in both structures A and B, supporting previous evidence showing that TM6 participates in the stabilization of CXCR4 homodimers[27,36]. In particular, mutations on TM6 like V242D[6.38] and L246P[6.42], both at the binding interface in dimeric structure B (Fig. 2B), were found by Isbilir et al. to inhibit the formation of homodimers and reduce the receptor basal activity[36]. Recently, higher-order structures of CXCR4 as nanoclusters[27] have been identified. The formation of such oligomers is inhibited by specific point mutations at Lys239[6.35], Val242[6.38], and Leu246[6.42] on TM6[27]. Interestingly, the same mutations do not hamper the formation of dimers[27], thus confirming that multiple CXCR4 binding interfaces are possible. We point out that all the mutated residues in that study (K239E[6.35], V242A[6.38], and L246A[6.42])[27] form stable interactions in dimeric structure B (Fig. 2B, inset), whereas they are only marginally involved in the binding mode of dimer A. Therefore, while the dimeric state B is affected by such mutations, the formation of the symmetric dimer A is still possible in the mutated receptor. This is confirmed by in silico mutagenesis experiments. In fact, introduction of mutations V242D[b 6.38] and L246P[b 6.42], or K239E[b 6.35], V242A[b 6.38], and L246A[b 6.42] in dimer B (Supplementary Fig. 7B on the right) destabilized the dimer with a $\Delta\Delta G$ of about 3–7 kcal/mol with respect to the wild-type structures, whereas their effect on state A was significantly lower (Supplementary Table 4).

Finally, we computed the $\Delta$SASA value of dimer A and dimer B with respect to the receptor monomeric state. The lower value of dimer A with respect to dimer B (16.59 nm² vs. 19.70 nm²) indicates a more compact structure for the latter. In dimeric structure A, the protomers are more exposed to the environment and more prone to interact with the surrounding phospholipids, cholesterol, and water molecules as discussed in the Supplementary discussion "The role of cholesterol in CCR5 and CXCR4 dimerization".

## CCR5–CXCR4 heterodimer

Similar to the previous systems, the CCR5–CXCR4 heterodimer has two lowest energy dimeric structures A and B with a symmetric and an asymmetric binding mode, respectively (Fig. 1C and Fig. 2C). The latter is energetically more stable than the former (−22.6 kcal/mol and −24.4 kcal/mol for A and B, respectively).

Dimeric structure A − In dimer A (see Source Data), both CCR5 and CXCR4 interact mainly through TM4 and TM5 (Fig. 2C, minimum A). On the extracellular side, a cluster of aromatic residues stabilizes the dimeric complex. This is made of Phe166[a 4.63], Phe182[a ECL2], Tyr184[a ECL2], Phe189[a 5.33], Trp190[a 5.34], Phe193[a 5.37], Phe260[a 6.60], Phe263[a 6.63], Phe264[a 6.64] and Phe172[b 4.61], Phe174[b 4.63], Trp195[b 5.34], Phe199[b 5.38]. At transmembrane level, a zip-like network of hydrophobic interactions engaged by Ile151[a 4.48], Val155[a 4.52] and Leu210[b 5.49], Ile213[b 5.52] is formed between the protomers (Fig. 2C and Supplementary Fig. 10A). At the intracellular side, polar contacts such as the H-bond between Lys138[a ICL2] and Asn143[b ICL2] contribute to further stabilize this dimeric structure.

Dimeric structure B− In dimeric structure B (see Source Data), TM1[a]–TM2[a]–H8[a] of CCR5 interact with TM6[b]–TM7[b] of CXCR4 (Fig. 2C,

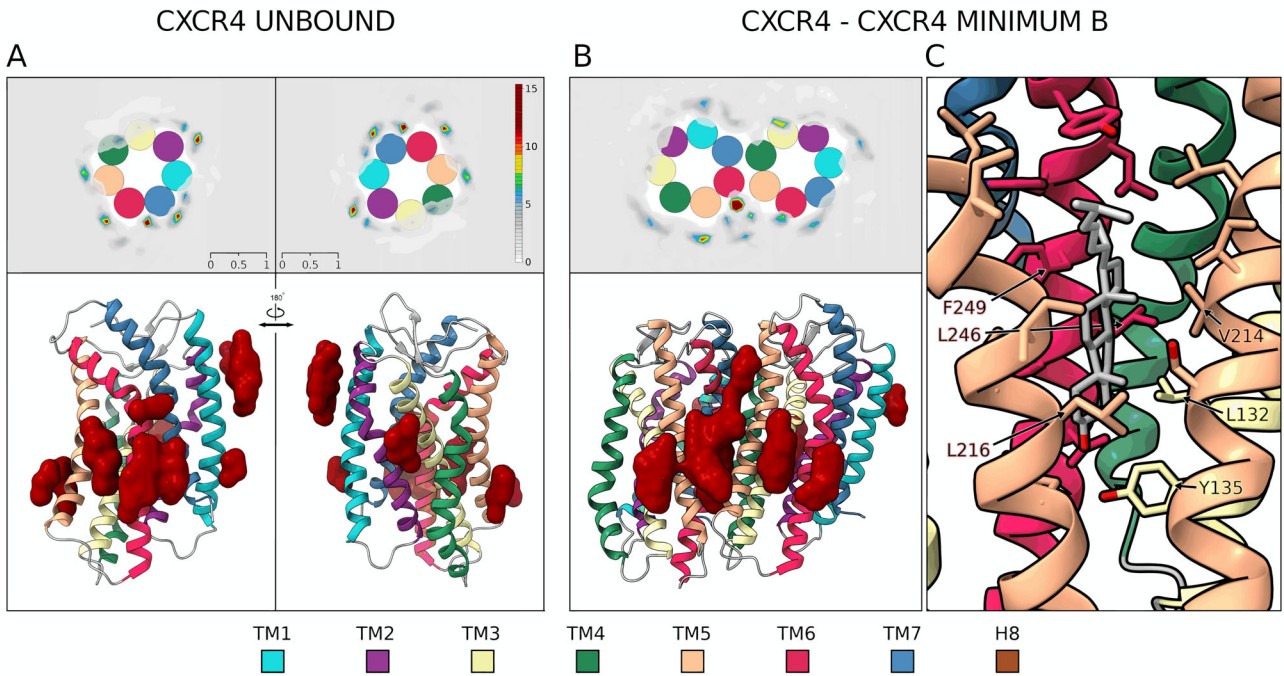

**Fig. 3 | Cholesterol molecules in CXCR4 homodimeric structure B. A** High-density regions of cholesterol molecules around the CXCR4 monomers are represented as 2D extracellular view (Top Left) and 3D atomistic structure (Bottom Left). **B** Additional high-density regions are present in both protomers, not at the dimer interface; some of those are also found in the monomeric state. **C** Atomistic detail of the cholesterol molecule stabilizing the CXCR4 dimeric structure B

represented as gray sticks together with its interacting residues. The dimeric structure B of CXCR4 hosts a binding site for cholesterol shaped by TM3-TM5 for one protomer and TM5-TM6 for the other, where polar and hydrophobic interactions are formed. Further discussion on the role of cholesterol in CXCR4 and CCR5 monomeric and dimeric forms is reported in the Supplementary Information. The color code of TM helices is the same reported in Fig. 1.

minimum B). We note that in this dimeric state, the helices at the binding interface are tilted, forming an angle of ~60°. At extracellular level, H-bonds and hydrophobic interactions are formed by polar and apolar residues such as Arg31$^{a\,1.33}$, Phe85$^{a\,2.59}$, His88$^{a\,2.62}$, Asp95$^{a\,ECL1}$, Phe96$^{a\,ECL1}$ and Asn35$^{b1.29}$, Phe36$^{b1.30}$, Phe40$^{b1.34}$, Lys282$^{b7.33}$, Trp283$^{b7.34}$ (Fig. 2C and Supplementary Fig. 10B). At transmembrane level, residues Ile42$^{a1.44}$, Phe45$^{a1.47}$, Val46$^{a1.48}$, Met49$^{a1.51}$, Leu50$^{a1.52}$ and Ile243$^{b6.39}$, Leu246$^{b\,6.42}$, Ala250$^{b\,6.46}$, Pro254$^{b\,6.50}$ establish a series of hydrophobic contacts. At the intracellular side, polar interactions are engaged by residues located on TM1-ICL2$^a$ and TM5-ICL1$^b$ like Asn57$^{a1.59}$, Arg60$^{a\,ICL1}$ and Ser224$^{b5.63}$, Y219$^{b\,5.58}$ while Phe311$^{a8.57}$, Phe312$^{a8.58}$, and Gln313$^{a8.59}$ on helix H8$^a$ interact with Lys236$^{b\,6.32}$, Lys239$^{b\,6.35}$, and Ile243$^{b\,6.39}$.

The ΔSASA values computed for dimer A and B relative to the monomeric state reveal that the lower energy dimeric structure A also has a lower ΔSASA estimate (ΔSASA dimer A = 12.65 nm;$^2$ ΔSASA dimer B = 15.80 nm$^2$). In addition, the CCR5−CXCR4 heterodimer A is the absolute lowest energy dimeric state compared to all the other dimeric structures identified in our study, and it also has the lowest ΔSASA value. These data indicate that the packing between protomers is not the main determining factor for the stability of GPCRs dimers, which is instead ruled by the type of interactions established between the protomers and the surrounding molecules like cholesterol.

### Effect of dimerization on CCR5 and CXCR4 functional mechanism

The fundamental biological question dealing with GPCRs dimerization is: do dimeric states affect receptor functioning? Previous studies have clarified the role of dimerization in GPCR activation[68]. However, the mechanism by which receptor dimerization can influence the activation process is still unclear. We decided to address this question by investigating the effect of the different dimeric states on the accessibility to the binding sites of the ligand (extracellular orthosteric binding site) and the G protein (intracellular).

Access to the orthosteric binding site − The access to the ligand binding site was studied by computing the volume of the binding cavity during the atomistic MD simulations on the monomeric, homo- and heterodimeric structures of CCR5 and CXCR4 (Fig. 4A–C). We found that in all the systems, the cavity volume ranges between 0.9 and 1.2 nm$^3$. These values are in line with those calculated for CCR5 bound to CCL5 (0.9 nm$^3$)[23] and gp120 (1.2 nm$^3$)[24]. The only exception is represented by CCR5 in the symmetric heterodimer A. Here, the volume of the ligand binding site is significantly reduced by ~25%. A closer inspection reveals that the CCR5−CXCR4 dimer A shows a closure motion of the CCR5's ECL2 over the binding cavity (Fig. 4G). This is due to the involvement at the dimer interface of TM4 and TM5− which are connected by ECL2−that induce such a motion of the loop. In addition, the extracellular ends of TM5 and TM6 are slightly shifted towards the binding site, further reducing the accessible volume for ligand binding (Fig. 4G). Prompted by a recent work showing that ligand binding to a minor pocket formed by TM2−TM3−TM7 in CXCR4 might reduce the formation of receptor homodimers[36], we also inspected the access to this alternative binding site. In particular, we found no significant rearrangement of this pocket in the homodimeric structures A and B. This finding was somehow expected since such a minor pocket is distant from the binding interfaces identified in our study. However, we cannot exclude that ligand binding to this pocket could trigger long-range allosteric effects that might affect receptor dimerization.

Access to G protein binding site − The access to the G protein binding site was evaluated by computing the distance between the intracellular ends of TM3 and TM6, defined as the distance between the Cα atoms of Arg126$^{3.50}$ and Arg232$^{6.32}$ for CCR5 and Arg134$^{3.50}$ and Lys236$^{6.32}$ for CXCR4 (Fig. 4D−F). In fact, TM6 is the helix mostly involved in the receptor large-scale conformational change occurring from the inactive to the active state[3], whereas TM3 is conformationally stable during receptor activation. As such, by looking at the change in

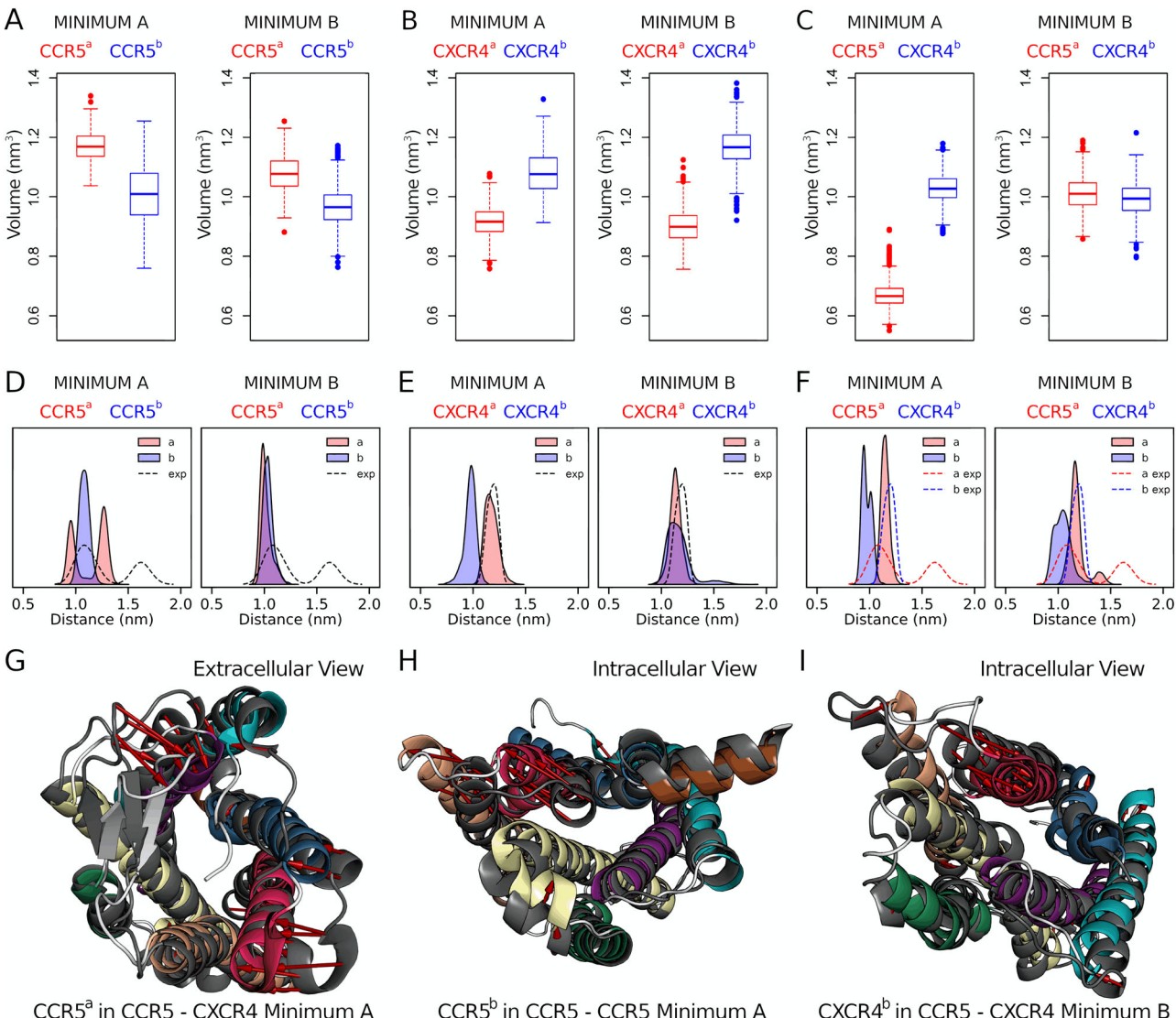

**Fig. 4 | Effect of dimerization on CCR5 and CXCR4 functional mechanism.**
Estimates of the volume of the ligand binding site for each dimeric structure.
**A** CCR5 homodimer. **B** CXCR4 homodimer. **C** CCR5–CXCR4 heterodimer. The
volume calculation was performed on the structures obtained from the atomistic MD simulations, using 500 frames for each system. The boxplots illustrate the
distribution of data, with the central box representing the interquartile range (IQR)
bounded by the first and the third quartile. The line inside the box denotes the
median, while the whiskers extend to the minimum and maximum values within 1.5
times the IQR. Data points beyond the whiskers are considered outliers and
represented as dots. Values of the distance between TM3 and TM6 for each dimeric
structure. **D** CCR5 homodimer. **E** CXCR4 homodimer. **F** CCR5–CXCR4 heterodimer.
The values were obtained from the atomistic MD simulations (4500 frames per
system). The same distances are calculated in the experimental structures
(11 structures for CCR5, 6 for CXCR4) and reported as kernel density estimation
(dashed lines). **G** Atomistic structure of CCR5 with reduced access to ligand binding
site. **H** Detail of CCR5 with TM5-TM6 in open conformation. **I** Close-up of CXCR4
with TM5-TM6 in the closed conformation. Here, the reference monomeric structures for CCR5 and CXCR4 are reported as gray cartoons, while displacements are
highlighted with red arrows.

the distance between these helices, it is possible to assess the activation state of a GPCR. A similar distance has also been used to define the
activation of the adenosine $A_{2A}$ GPCR[69]. Looking at Fig. 4D, in the CCR5
homodimeric structure A (i.e., symmetric binding mode), TM6
assumes a more open conformation, shifting the receptor toward its
active form (TM3-TM6 distance of 1.6 nm)[70], which might favor the G
protein binding and in turn the activation of the signaling cascade
(Fig. 4D, left and Fig. 4H). Conversely, in the other CCR5 homodimeric
structure B (Fig. 4D, blue curve) and in both the CXCR4 homodimeric
structures (Fig. 4E), there are no major structural changes that could
facilitate the binding of the G protein with respect to the experimental
inactive state. The most striking result was found in the CCR5–CXCR4
heterodimer. Here, in both dimeric structures A and B, CXCR4 assumes
a much closer conformation that hampers the G protein binding, thus

locking this protomer in an inactive conformation (Fig. 4F, blue curves,
and Fig. 4I). On the other hand, the G protein binding site in CCR5 has
no significant alteration if compared with the experimental inactive
structures (Fig. 4F, red curves).

Taken together, our findings indicate that dimer formation can
induce one of the two protomers to assume a certain conformation
with a specific affinity for the ligand and the G protein. Therefore,
receptor dimerization *de facto* represents a fine allosteric mechanism to modulate GPCR activity, as also proposed for other
receptors[71]. In this paradigm, disclosing GPCRs dimeric structures at
atomistic resolution is of paramount importance to elucidate,
understand, and possibly exploit receptor dimerization for an exogenous regulation of the receptor activity and, in turn, the signaling
cascade.

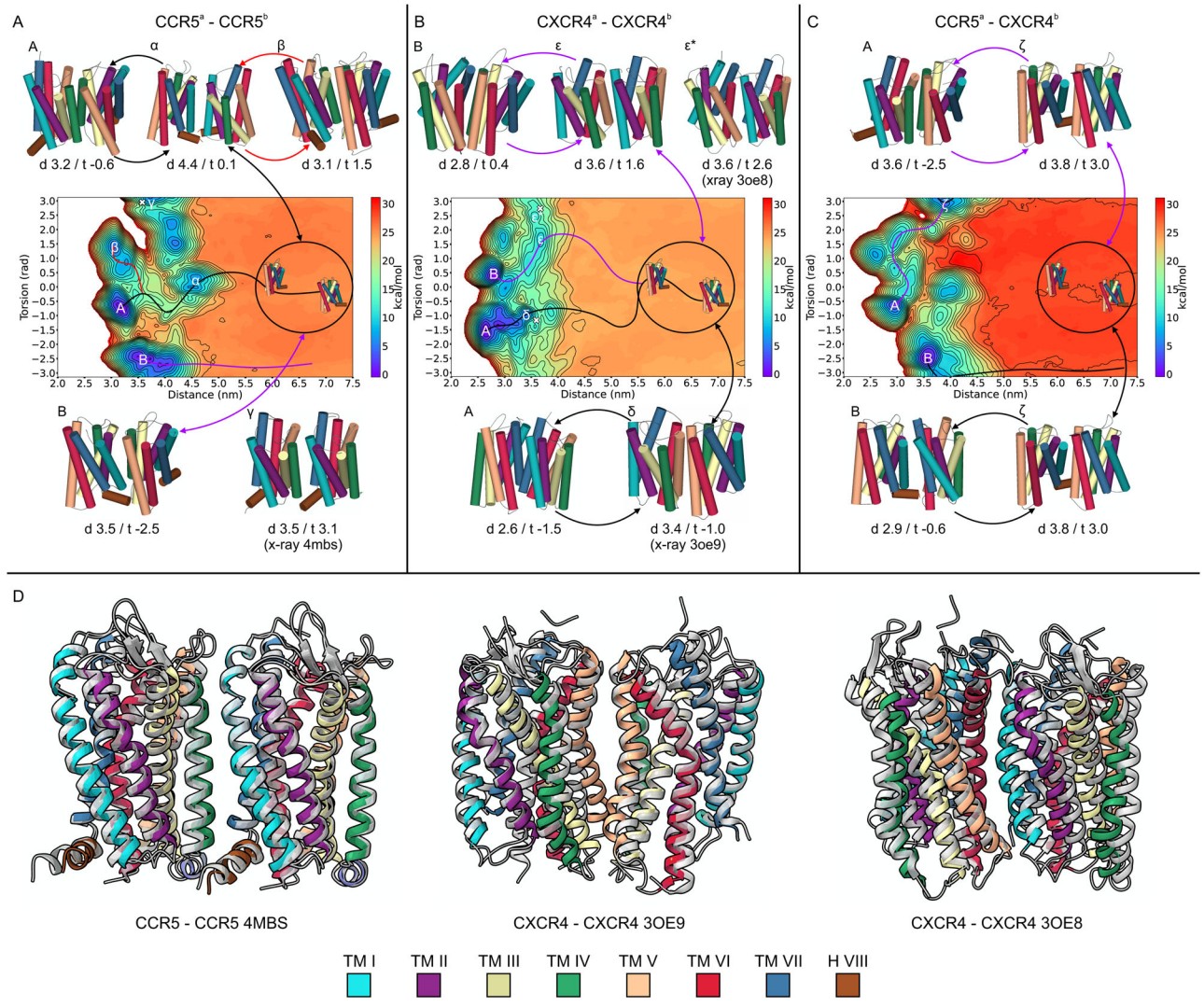

**Fig. 5 | CCR5 and CXCR4 dimerization mechanism.** Representation of the BFES and the structures of the metastable states and minima identified for each system. **A** CCR5–CCR5. **B** CXCR4–CXCR4. **C** CCR5–CXCR4. In the case of CCR5–CXCR4 heterodimer, CCR5 is reported on the left, whereas CXCR4 is on the right. The lowest energy basins of each system on the BFES are highlighted by the A and B white letters, whereas the position of the metastable states is reported using white Greek letters. White crosses highlight the position on the BFES of the experimentally resolved CCR5 and CXCR4 homodimers. For each BFES, the lowest energy paths (LEPs) computed from the MetaD–CG calculations starting from each minimum or metastable state are reported as black, red, or purple solid lines. The transitions between metastable states and energy minima identified by the LEP are represented using arrows of matching colors connecting the structures depicted in cartoons on the top and bottom of each BFES. **D** Comparison of metastable dimeric states identified by CG-MetaD with the following X-ray structures: (Left) CCR5 dimer in 4MBS[19] (RMSD 0.17 nm); (Center) CXCR4 dimer in 3OE9[21] (RMSD 0.30 nm); (Right) CXCR4 trimer in 3OE8[21] (chain A and B) (RMSD 0.29 nm). RMSD was computed on the Cα atoms of TM helices. Here, the bulky crystallization adjuvant molecules bound to the intracellular part of CCR5 and CXCR4 are omitted for clarity, while they are displayed in Supplementary Fig. 12. Experimental structures are represented as gray cartoons, whereas the CG-MetaD structures are displayed as color coded cartoons.

## Dimerization mechanism

In our study, hundreds of (un)binding events between the receptors were reproduced, with the GPCRs going back and forth from the monomeric (unbound) to the dimeric (bound) states. Therefore, our simulations not only provided the structures of the energetically more stable states but also yielded insight into the dimerization mechanism. To this scope, we calculated the lowest energy paths (LEPs) of receptor dimerization and identified the functionally relevant metastable states (see SI Methods for details). We report the LEPs and the identified metastable states in Fig. 5A–C.

CCR5 dimers − In the case of CCR5 homodimers, we identified two LEPs from the monomeric to the dimeric states A and B (solid lines in Fig. 5A). The LEP ending in B directly reaches the dimer basin, while the LEP leading to A passes through the metastable state α, which shows the symmetric binding interface TM1–TM2–H8. Then, it splits

into two paths, one leading to A and the other reaching a dimeric state, β, at a higher energy value than A and B (see Supplementary Information for details). Furthermore, it is worth noting the metastable state γ, which represents an intermediate state along a higher energy path connecting the metastable state α to minimum B. Such state has an asymmetric binding interface formed by TM4/TM1–TM7–H8 and is very similar to the experimental dimeric structure of CCR5 resolved in presence of the inhibitor maraviroc (PDB ID 4MBS), with a low Root Mean Square Deviation (RMSD) value of 0.17 nm (Fig. 5D)[19]. In order to further assess energetically the experimental 4MBS structure, 160 μs of CG-MD calculations were performed using such structure as a starting state. The results show that the X-ray starting pose is left in favor of closer, energetically more stable dimeric structures with an average RMSD between the starting and the final structures of 0.43 nm (Supplementary Fig. 11). This confirms that the experimental 4MBS

structure represents a metastable dimeric state of CCR5. In this regard, it is worth noting that recent works propose that the binding of maraviroc to CCR5 induces a specific receptor conformational change stabilizing such a dimeric state[13,32].

CXCR4 dimers − One single LEP was identified for the CXCR4 homodimer that, from the monomeric state, splits in two paths reaching the two lowest energy dimers A and B. One LEP passes through metastable state δ and then reaches A, whereas the other passes through metastable state ε before reaching B (black and purple solid lines in Fig. 5B, respectively) (see Supplementary Information for details). Interestingly, the metastable state δ shows the symmetric binding interface TM5-TM6, similar to that found in the experimental CXCR4 structures with PDBIDs 3OE9, 3OE8, and 3ODU (RMSD 0.30 nm, Fig. 5D)[21]. On the other hand, a metastable state close to ε has the asymmetric binding interface TM5$^a$–TM6$^a$/TM1$^b$ (ε* in Fig. 5B), similar to that present in the CXCR4 trimeric X-ray structure with PDBID 3OE8 (RMSD 0.29 nm, Fig. 5D)[21]. As previously done for the CCR5 dimeric structure 4MBS, we assessed the structural stability of the dimers 3OE9 and 3OE8, performing 320 µs of CG−MD calculations (160 µs for each system). In both cases, the system leaves the starting structure to reach a closer, energetically more stable dimeric state (see Supplementary Fig. 11 and Supplementary Information for details). Interestingly, in the case of 3OE9, the system lands in the lowest energy minimum A since no significantly high energy barrier separates the starting X-ray structure from the lowest energy one.

CCR5−CXCR4 dimers − One single LEP connects the monomeric states and the dimers A and B. This path passes through the metastable state ζ with a binding interface formed by TM1$^a$–TM7$^a$–H8$^a$/TM5$^b$–TM6$^b$. From this state, the system can reach A and B following two separate paths (purple and black solid lines in Fig. 5C, respectively).

Overall, our results suggest a multi-step mechanism of dimerization with the presence of lowest energy and metastable dimeric states, in line with what was proposed by other colleagues[72–76]. We found that the CCR5 and CXCR4 X-ray dimeric structures represent metastable states. In this regard, we point out that in these complexes, the receptors are bound at the intracellular level to bulky crystallization adjuvant molecules−rubredoxin and lysozyme for CCR5 and CXCR4, respectively−which impede by steric hindrance a closer contact between the GPCRs, stabilizing them in metastable states without reaching the lowest energy minima (see Supplementary Fig. 12).

## Effect of membrane lipid composition and G Protein coupling on receptor dimerization

Lipids effect − Lipids are known to play an important role in protein-protein binding interaction, as seen for several GPCRs, including chemokine receptors[47,50,51,77,78]. In order to investigate the effects of different lipid compositions on the CCR5 and CXCR4 dimers, we performed over 300 µs CG−MD calculations on the six identified dimeric structures in a realistic plasma membrane model (Fig. 6A). This is asymmetrically composed of ten differently saturated phospholipids and cholesterol concentration at 25%, mimicking the composition of in vivo cell membrane[47]. The list of the components of the plasma membrane model is reported in Supplementary Table 3A and in the top and bottom cake diagrams of Fig. 6A. The results show that all the six dimeric structures are stable in the plasma membrane model with the sampling confined to each energy minimum (Fig. 6B−D). However, in CCR5 (Fig. 6B) and CCR5−CXCR4 dimers A (Fig. 6D) the minimum CV values are slightly increased. This is due to the presence of a DOPE and cholesterol molecule at the dimer interface that mediate the binding interaction between protomers without changing the original binding interface. Therefore, our results in plasma membrane confirm the dimeric states identified using the POPC/cholesterol membrane model and show that certain lipids, like DOPE and cholesterol, might interplay with protomers during dimerization.

The structural stability of the six dimers identified for CCR5 and CXCR4 has been further assessed in the plasma membrane model by means of 12 µs atomistic MD calculations. During these simulations, the binding mode between receptors in all the dimeric structures is stable with a low average RMSD of 0.10 nm computed for the receptors backbone atoms (see Supplementary Fig. 4). Furthermore, additional unbiased CG−MD and CG−MetaD calculations on CCR5−CCR5 binding in the plasma membrane model were performed using the latest version of the Martini force field (Martini 3)[79], which was released during the review process of the present article. As can be seen from Supplementary Fig. 13, after only 30 µs of simulations, the phase space exploration by means of CG−MetaD remains remarkably superior if compared with that of unbiased CG simulations, including the BFES region of the low energy minima. In this direction, it would be interesting to investigate in the future the effects of different plasma membrane models with diverse lipid composition on receptor diffusion and interaction in membranes during dimerization and oligomerization.

G protein effect − GPCR activation entails a conformational rearrangement of TMs 5–7 that allows access to the intracellular G protein binding site[3]. This conformational change might affect the quaternary structure of the dimers due to the different steric hindrance of the GPCR in the activated form, especially when coupled with a G protein. In order to investigate such an effect on the dimers identified in our study, we performed over 200 µs CG−MD calculations on the CCR5 dimers A and B and the CCR5−CXCR4 heterodimers A and B with CCR5 in the active state−i.e., with the agonist Chemokine C−C Motif Ligand 3 (CCL3) bound to the orthosteric binding site and the Gαβγ protein heterotrimer coupled to the intracellular binding site, henceforth defined as aCCR5(G) (see "Methods" for details). We note that we limit our study to CCR5 as no active experimental structure of CXCR4 coupled with a G protein has been available so far. The simulations performed in the plasma membrane model previously introduced show that in both systems (aCCR5(G)−CCR5 and aCCR5(G)−CXCR4), the dimeric structure A is preserved with the protomers slightly more distant due to the presence of the G protein (Fig. 7). These data agree with the previously discussed atomistic MD results showing CCR5 in a more open state in homodimer A (see Effect of dimerization on CCR5 and CXCR4 functional mechanism paragraph) and confirm that dimers using TM4−TM5 as binding interface are prone to be activated (Figs. 7B−E). On the other hand, the dimeric structures B in both aCCR5(G)−CCR5 and aCCR5(G)−CXCR4 change (Fig. 7C, F). In particular, in aCCR5(G)−CCR5, the protomers slightly rotate, interacting through TM4−TM5/TM1−TM7−H8 instead of TM4−TM5/TM1−TM2−H8 found in the homodimer formed by two inactive CCR5 molecules (Fig. 7C). In aCCR5(G)−CXCR4, the effect of G protein on receptors binding is stronger: the TM1−TM2−H8/TM6−TM7 interface identified in the inactive CCR5−CXCR4 dimer is left in favor of TM4−TM5/TM1−TM2−TM3 (Fig. 7F). Finally, it is worth noting that the binding interface TM1−TM2−TM8 is not used by the active CCR5 (aCCR5(G)), which instead binds to the other protomer− either CCR5 or CXCR4−always through TM4−TM5 (Fig. 7C, F on the left), suggesting a dimerization interface selection mechanism mediated by the G protein.

## Discussion

In the present work, we investigated the dimerization mechanism of the chemokine receptors CCR5 and CXCR4 employing the advanced free-energy technique CG-MetaD. We reproduced minute timescale binding events between the GPCR protomers that allowed providing a holistic picture of the dimerization process (see Supplementary Movie 1). Interestingly, for each of the investigated systems (CCR5 homodimer, CXCR4 homodimer, and CCR5−CXCR4 heterodimer), two lowest energy dimeric structures−hence most probable−were disclosed, one characterized by a symmetric binding mode A (i.e., the two

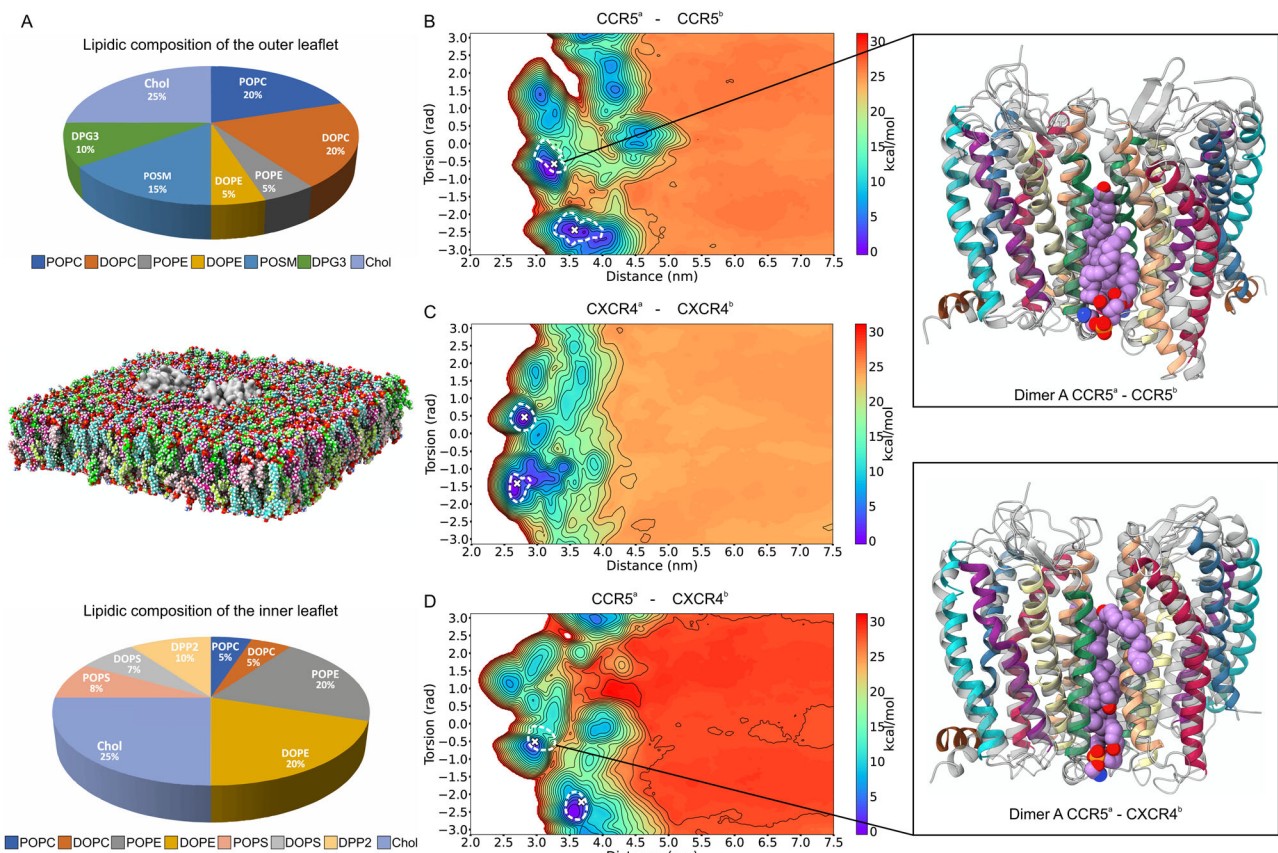

**Fig. 6 | CCR5 and CXCR4 dimers in plasma membrane model. A** Composition of the plasma membrane model with details of the outer (top) and inner (bottom) leaflets and representation of the proteins in the membrane (center). The proteins are shown as silver surfaces, whereas the lipids are colored in lime (POPC = 1-palmitoyl-2-oleoyl-sn-glycero-3-phosphatidylcholines), dark green (POPS = 1-hexadecanoyl-2-(9Z-octadecenoyl)-sn-glycero-3-phosphoserine), green-yellow (POPE = 1-palmitoyl-2-oleoyl-sn-glycero-3-phosphoethanolamine), turquoise (Chol = cholesterol), medium violet red (DOPC = 1,2-dioleoyl-sn-glycero-3-phosphatidylcholines), light coral (DOPS = 1,2-di-(9Z-octadecenoyl)-sn-glycero-3-phosphoserine), pink (DOPE = 1,2-dioleoyl-sn-glycero-3-phosphoethanolamine), cyan (DPP2 = CG model corresponding to the atomistic C16:0 dipalmitoyl phosphatidylinositol 4,5-bisphosphate (DP-PIP2)), green (POSM = N-(9Z-octadecenoyl)-hexadecasphing-4-enine-1-phosphocholine), and chartreuse (DPG3 = neuAcalpha2-3Galbeta1-4Glcbeta-Cer(d16:1/16:0)). The hydrogen atoms of the lipids are represented as white spheres, whereas oxygens are depicted in red and nitrogens in blue.

At the center, exploration of the BFES starting from the lowest energy dimers of each system embedded in an in vivo-mimicking cell membrane model. **B** CCR5–CCR5. **C** CXCR4–CXCR4. **D** CCR5–CXCR4. The white dashed lines represent the region of the BFES sampled by each system during 50 μs of CG–MD simulations. The starting points of each calculation are highlighted by white crosses. In the background, the dimerization BFES for each system computed via CG−MetaD calculations in the POPC/cholesterol 9:1 bilayer is shown. On the right, the insets show the comparison between the quaternary structures of the CCR5–CCR5 and CCR5−CXCR4 dimers A in the plasma membrane model (depicted in color-coded cartoons) and in the POPC/cholesterol membrane model (gray cartoons). DOPE and cholesterol molecules interposed between the protomers of the dimers simulated in the plasma membrane model are represented as spheres with carbons colored in purple, nitrogens in blue, oxygens in red, and sulfur atoms in orange.

protomers interact through the same helices) and the other by an asymmetric binding mode B (Fig. 1 and Fig. 8). Our findings indicate that some helices are more prone to form dimers with respect to the others, in agreement with experimental data[14,21,37,67]. For instance, TM4 and TM5 helices are involved in the binding mode of both the dimeric structures of CCR5, the dimeric structure B of CXCR4 and the dimeric structure A of CCR5−CXCR4. In addition, TM4−TM5 is also the preferred interface of the active form of CCR5−complexed with G protein −when binding to another protomer, either CCR5 or CXCR4, suggesting a dimerization interface selection mechanism mediated by the G protein (Fig. 7A, D).

The dimeric structures identified in our study show that at the extracellular level, aromatic residues like phenylalanine and tryptophan are typically the inter-protomer interacting residues, integrated by stronger interactions established by polar and charged residues of the ECLs. In the transmembrane domain, the binding between the protomers is stabilized by a series of hydrophobic contacts engaged by residues like leucines, isoleucines, and alanines, forming a zip-like

interaction motif. On the intracellular side, polar and charged residues of facing protomers usually interact, further stabilizing the dimeric structure (Fig. 2). A more quantitative analysis is provided by our in silico mutagenesis experiments in which we estimated the effect on the binding energy for the dimeric structures identified in our study of aminoacidic mutations reported in the literature to affect receptor dimerization (see Supplementary Table 4). Among these, L196K[5.40], I200K[5.44], and L205K[5.49] are disruptive on both the dimeric structures A and B of CCR5, whereas K239E[6.35] and L246P[6.42] affect both the dimeric structures A and B of CXCR4. Furthermore, we mutated into alanine (Ala scan) each residue closer than 8 Å to the binding interface in the CCR5 and CXCR4 dimeric structures. Three tryptophan residues at the binding interface of both homo- and heterodimers—i.e., Trp190[5.34] (CCR5), Trp283[7.34], and Trp195[5.34] (CXCR4)—were found to significantly contribute to the energetic stability of the dimeric structures (see Supplementary Table 5). Our results prompt to further investigate their role in CCR5 and CXCR4 dimerization using molecular simulations, possibly in combination with spectroscopic experiments, such

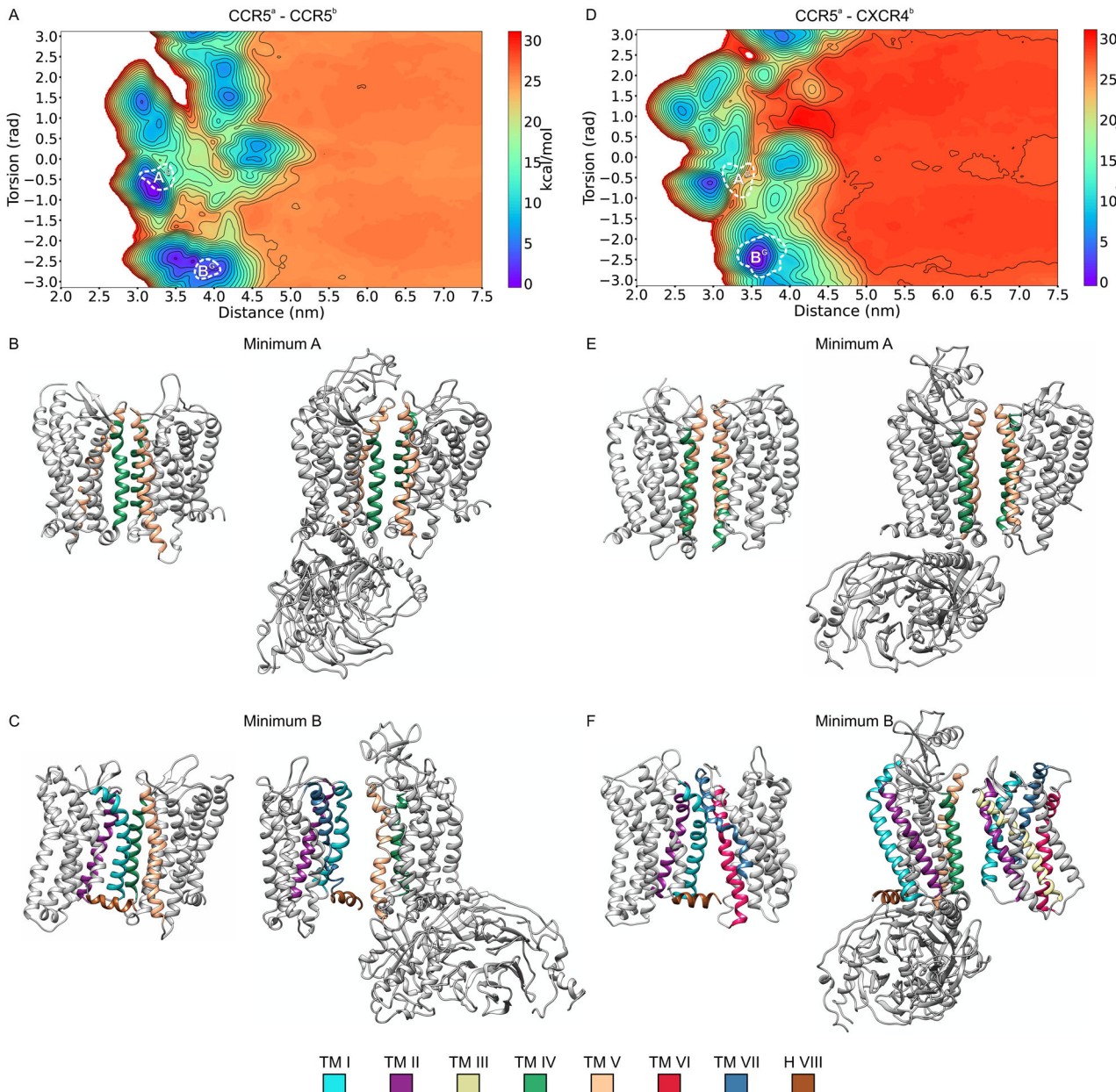

**Fig. 7 | Effect of G protein on the energy minima.** Exploration of the BFES starting from the lowest energy dimers containing an active CCR5 receptor and an inactive CCR5/CXCR4 protein embedded in the realistic plasma membrane model. aCCR5(G) represents an activated CCR5 protein bound to CCL3 and the Gαβγ heterotrimer. **A** aCCR5(G)−CCR5. **D** aCCR5(G)−CXCR4. The white dashed lines represent the boundaries of the CV regions explored by each system during over 50 μs of CG−MD simulations. These regions are denoted as A^G and B^G to distinguish them from the minima identified for the inactive receptors in the absence of G protein. The back-mapped centroids of the A and B minima extracted after clusterization of the aCCR5(G)−CCR5 trajectories are represented as color-coded cartoons, respectively, on the right of (**B**, **C**). On the left of each panel, the corresponding centroids obtained from the initial MetaD−CG calculations with the inactive receptors are reported. Similarly, the backmapped centroids of the A and B minima extracted after clusterization of the aCCR5(G)−CXCR4 trajectories are represented as color-coded cartoons on the right of (**E**, **F**). On the left of each panel, the corresponding centroids obtained from the initial MetaD−CG calculations with the inactive receptors are reported.

as those employing 5−13C-methyl-deutero-tryptophan TROSY−NMR technology, capable of detecting conformational changes of tryptophan residues during GPCR dimer formation.

The identified dimeric structures differ in the access to the binding sites of the ligand and the effector G protein. In particular, in CCR5 dimeric structure A the G protein binding site of one protomer is more open, stabilizing the receptor in an activable state (Fig. 4D). This is further confirmed by our results obtained using the active form of CCR5, aCCR5(G), where such dimeric structure is also found in the presence of the Gαβγ heterotrimer (Fig. 7). On the other hand, in

CXCR4 dimeric structure A and in both the CCR5−CXCR4 homodimeric structures A and B, the G protein binding site of CXCR4 is shifted towards the closed (inactive) state (Fig. 4E). Such findings might open interesting opportunities in drug design studies. In fact, developing bivalent ligands capable of binding at the same time CCR5 and CXCR4 in the diverse dimeric structures might result in a fine allosteric modulation of these receptors. For instance, ligands stabilizing CCR5 homodimeric structure A might prolong the activation state of the receptor, thus enhancing CCR5-related cell signaling. On the other hand, ligands capable of stabilizing CCR5−CXCR4

**Fig. 8 | Artist's impression of homo- and heterodimerization mechanism of CCR5 and CXCR4.** CCR5 and CXCR4 can assume three different free-energy states: (i) monomeric states; (ii) metastable states; and (iii) lowest energy states. From the monomeric state, CCR5 and CXCR4 can form metastable states, and by crossing relatively small energy barriers, the receptor can reach the lowest energy dimeric states that are characterized by a symmetric and asymmetric binding modes. The stability of the states might be influenced by specific cell conditions, including the presence of ligands and membrane components like cholesterol molecules, phospholipids, or other membrane proteins.

heterodimeric structures A and B might dampen all the cellular pathways regulated by CXCR4.

Our findings indicate that dimerization *de facto* is a fine allosteric modulatory mechanism of GPCR activity—alternative to ligand-based allosteric modulation—in which the activity of one protomer can be modulated by the binding of another protomer (crosstalk-regulation)[71,75], and a certain dimeric state might selectively favor or disfavor the binding of G proteins, beta-arrestins, or other effector proteins, eventually triggering a specific signal cascade. However, we have shown that the CCR5 and CXCR4 dimerization mechanism is a rather dynamical process in which a number of alternative, energetically metastable dimeric states might coexist, as also reported for other GPCRs (Fig. 8)[6,36,37,50,73,76,80]. As a consequence, specific cell conditions or presence of ligands and lipids might stabilize one dimeric state with respect to the others and, in turn, modulate receptor activity. This is the case of CCR5 and CCR5–CXCR4 dimers A, wherein the plasma membrane model, a DOPE phospholipid and cholesterol molecule mediate the binding interaction between protomers (Fig. 6).

Our results also prompt some considerations regarding the possibility of forming receptor oligomers starting from the dimers. In particular, both CCR5 and CXCR4 have a symmetric and asymmetric binding mode. Therefore, either dimeric structure has the possibility to form trimers, tetramers, and even oligomers by using the free binding interface to complex with another protomer. For instance, TM4–TM5 is involved in both the CCR5 dimeric structures A and B, whereas TM1–TM2–TM8 only in the dimeric structure B (Fig. 2A). Thus, in both cases, there is one binding interface, either TM4–TM5 or TM1–TM2–TM8, available for binding another protomer. The same applies to CXCR4, in which TM6–TM7 forms the binding site in both the symmetric and asymmetric dimeric structures A and B, whereas TM4–TM5 is involved only in the dimeric structure B (Fig. 2B). This observation assumes even more relevance in light of the works of Martínez-Muñoz et al.[27]. and Işbilir et al.[36] showing that the CXCR4 triple mutant K239E$^{6.35}$–V242A$^{6.38}$–L246A$^{6.42}$ on TM6 does not form receptor oligomers, while it is still able to form dimers. In fact, such residues are involved in the CXCR4 asymmetric dimeric structure B, while they are only marginally involved in the symmetric dimer A. As a result, such mutations disrupt the possibility for CXCR4 to bind another protomer through the binding mode of dimer B, while the mutated receptor still retains the ability to form dimers via the symmetric binding mode A, as confirmed by our in silico mutagenesis data (see Supplementary Table 4 and Supplementary Fig. 7B). However, a deeper characterization of the oligomer forms of these chemokine receptors is required and represents a major challenge in the near future that could benefit from combining cutting-edge techniques like free-energy CG–MetaD calculations and super-resolution imaging to resolve higher-order GPCRs structures.

Previous computational studies proposed CCR5 and CXCR4 dimeric structures[50,51] that are worth comparing with those identified in our work. In particular, Pluhackova et al. showed that CXCR4 can

form asymmetric dimers through TM1/TM5-7, whereas cholesterol might stabilize the symmetric TM3–TM4 dimer interface[50]. While such dimeric states are found in our study (Fig. 5), they turn out to be at higher energy values compared to the symmetric TM1–TM6–TM7 and the asymmetric TM6–TM7/TM5–TM5 dimer interface of structures A and B, respectively. In a second work, the same authors investigated CCR5 and CCR5–CXCR4 dimerization in a POPC membrane with 0% and 30% of cholesterol[51]. Here, the most populated dimer interfaces are asymmetric, where CCR5 forms homodimers through TM1–H8/TM4–TM5 and TM1–H8/TM5–TM6–TM7, whereas CCR5 and CXCR4 interact through TM1 and TM4–TM5–TM6–TM7. At variance with these results, we show that the lowest energy CCR5 and CCR5–CXCR4 dimers have symmetric binding interface composed of TM4–TM5. However, low energy asymmetric dimers are also possible through TM1–TM2–H8/TM4–TM5 and TM1–TM2–H8/TM6–TM7 for CCR5 and CCR5–CXCR4 dimers, respectively (Fig. 5). The different results might be attributed to the different cholesterol concentration of the membrane and the diverse simulation techniques employed. In fact, both in ref. 50,51, the authors performed hundreds of standard coarse-grained molecular dynamics simulations without, however, reaching a full exploration of the dimerization-free energy landscape. Differently, here we have combined metadynamics with coarse-grained molecular dynamics. This allows for enhancing the sampling of the phase space and achieving a converged dimerization free-energy landscape. In doing so, each possible dimeric state is visited and energetically evaluated—including those proposed in ref. 50,51—providing a thorough description of the receptor dimerization mechanism. Proof of that is further given by the fact that the X-ray dimeric structure of CCR5 (PDB ID 4MBS) is identified in our study (see Fig. 5), while it is not found by standard CG–MD simulations reported in ref. 51.

In conclusion, our results offer unprecedented structural insights into the dimerization mechanism of CCR5 and CXCR4 and of GPCRs more in general, considering the conservative nature of the functional mechanism within this receptor family. The dimeric structures resolved at atomistic resolution in our study (PDB files available in Supplementary Materials and at www.pdbdb.com) open so far unexplored routes for the regulation of the activity of these chemokine receptors through the structure-based design of ligands capable of modulating the formation of dimers, with therapeutic potential in the fight against HIV, cancer, and immune-inflammatory diseases related to these chemokine receptors.

## Methods
### Systems setup
The starting conformation of the CCR5 and CXCR4 protein has been taken respectively from the 4MBS[19] and 3OE9[21] X-ray structures. The structures have missing residues that were modeled with the MODELLER software[81]. The atomistic structures were first converted to the MARTINI 2 force-field[54] using the Martinize v2.5 tool[82] and the Elastic Network in Dynamics (ELNEDYN)[83] representation to retain the

proteins' secondary and tertiary structures. In the next step, two protomers were placed at a distance of about 7 nm and inserted in a squared CG phospholipid bilayer with a side of 20 nm composed of 1-palmitoyl-2-oleoyl-sn-glycero-3-phosphocholine (POPC)/cholesterol with 9:1 ratio. This system was then subjected to 10,000 steps of steepest descent minimization, followed by a multistage equilibration protocol detailed in Supplementary Methods.

## CG–MetaD simulations

The CG–MetaD simulation was performed at 300 K with an integration time step of 20 fs, using the well-tempered version of metadynamics[84] implemented in Plumed 2.3[48]. Ten parallel simulations were performed with GROMACS v. 5.1[85] according to the multiple walker[63] (MW) approach, each walker started from the equilibrated conformation of the system. The distance between the two proteins ($r$) and a torsion that describes the reciprocal orientation ($\Omega$) were chosen as collective variables (CVs). Details about the CV definition can be found in Supplementary Fig. 1. Gaussians of height 0.5 kJ/mol and width 0.04 nm for $r$ and 0.06 rad for $\Omega$ were used and deposited every 5000 steps with a bias factor of 20 for each walker. An upper wall limit was set for the $r$ CV at 8 nm to limit the exploration of unbound states (Supplementary Fig. 14). Details about parameters for CG–MetaD simulations are reported in Supplementary Methods.

The binding free energy $\Delta G^0_{bind}$ between the protomers was calculated as in refs. 63,84 using the following formula:

$$K_{bind} = \frac{\int_{bound} dr\, e^{-\beta\, PMF(r)} 2\pi r_{unb}}{e^{-\beta\, PMF(r_{unb})}} \quad (1)$$

$$\Delta G^0_{bind} = -k_b T \ln(K_{bind} C_0) \quad (2)$$

Here, $K_{bind}$ is the binding constant; $\beta$ is equal to $(k_b T)^{-1}$, where $k_b T$ at 300 K for a mole of compounds corresponds to 0.596 kcal mol$^{-1}$; $PMF(r)$ is the potential of mean force as a function of the $r$ CV obtained from CG–MetaD calculations; $r_{unb}$ is the reference distance for the unbound state; $2\pi r_{unb}$ is the entropic correction accounting for the motion of the proteins in the $xy$ plane of the membrane; and $C_0$ is the standard concentration of 1 M for all reacting molecules, useful for comparison with experiments.

## CG–MD simulations

The lowest energy dimeric structures identified by CG–MetaD were clustered with the GROMOS clustering method[85], based on the transmembrane backbone beads, and a distance cut-off of 0.2 nm. Additional 50μs of refinement CG–MD simulations using GROMACS v.5.1[85] were performed starting from the centroids of the most populated clusters for each system, following 10,000 steps of steepest descent minimization and 250,000 steps at 4 fs timestep. Details about parameters for CG–MD simulations are reported in Supplementary Methods.

## AT–MD simulations

The CG dimeric structures coming from the refinement CG–MD simulations were converted to atomistic structures using a backmapping protocol[49] with the CHARMM36 force-field[86]. In detail, the dimers were backmapped to their atomistic counterparts using the backward.py script provided by Wassenaar et al.[49]. These structures have been further refined via multiple minimization and equilibration using the initram-v5.sh script provided by Wassenaar et al.[49]. The systems were then subjected to 20,000 steps of steepest descent minimization, followed by an equilibration stage of 3 ns, in which timestep was increased and restrains were gradually decreased (details in Supplementary Methods). Finally, production runs for a total simulation time of 3 μs were performed to obtain the final atomistic structures

(the details about parameters for AT–MD simulations can be found in Supplementary Methods).

During the review process, an additional 12 μs AT–MD calculations on the CCR5 and CXCR4 dimeric structures identified by CG–MetaD were performed using a realistic plasma membrane model whose composition is reported in Supplementary Table 3C. The CG dimeric structures were converted to atomistic level using the CHARMM-GUI webserver[87].

## Structural characterization of CCR5 and CXCR4 dimers

For each dimeric structure, the contact area at the binding interface was calculated as the difference in the solvent-accessible surface area between the monomeric and the dimeric states (ΔSASA). This estimate was computed using the PISA server[88], whereas the per-residue contribution to dimerization was computed using the POPSCOMP server[89]. The ΔSASA values reported in Supplementary Figs. 5, 8, and 10 were computed using the gmx sasa tool of GROMACS[85] and all data were analyzed and plotted with R. Molecular graphics were obtained using Pymol, UCSF Chimera[90], and UCSF ChimeraX[91].

In order to analyze the distribution of cholesterol molecules around the proteins, states belonging to each minimum and to the unbound state ($r$ CV higher than 6.5 nm) were extracted, proteins were aligned based on backbone beads, and the positions of cholesterols centers of mass were collected. Densities for cholesterol molecules were computed in 2D and 3D using grid spacing 0.15 nm and by normalizing to the value of 1 the average density of cholesterol in the bulk membrane.

The volume of the GPCR ligand binding cavity was estimated using an in-house script inspired by POVME[92] that computes accessible points on a predefined grid, as explained in previous works[93,94]. The grid size was defined based on the starting conformation, including a tolerance to account for the conformational changes of the protein during the simulation. In all the systems, the receptors were aligned on the Cα atoms of TM helices and the same grid was used.

The activation states of CCR5 and CXCR4 were assessed by computing the distance distribution between TM6 (residue Arg232[6.32] for CCR5 and Lys236[6.32] for CXCR4) and TM3 (residue Arg126[3.50] for CCR5 and Arg134[3.50] for CXCR4) that is a widely used hallmark of the receptor conformational change allowing the G protein binding and in turn its activation. A similar distance has also been used to define A2A GPCR activation[69]. We note that Arg[3.50] was purposely chosen due to its involvement in the ionic lock typically present in the inactive states of GPCRs, and Arg[3.50] and Arg/Lys[6.32] have a high degree of conservation in class A GPCRs, thus allowing further comparisons with other receptors[69]. The experimental values related to the opening of the G protein binding site were calculated using the crystallographic and cryoEM structures available for CCR5 (PDB IDs 4MBS[19], 5UIW[23], 6AKX[25], 6AKY[25], 6MEO[24], 6MET[24], 7O7F[70], 7F1Q[95], 7F1R[95], 7F1S[95], 7F1T)[95] and CXCR4 (PDB IDs 3ODU[21], 3OE0[21], 3OE6[21], 3OE8[21], 3OE9[21], 4RWS)[22], both in the inactive and, when available (CCR5 only), active states. Their distributions were computed using the kernel density estimation (KDE) approach with a bandwidth of 1 (CXCR4) and 0.5 (CCR5). The distribution of the distance values computed from the AA-MD calculations was obtained by using KDE with a bandwidth of 1.

## Reporting summary

Further information on research design is available in the Nature Portfolio Reporting Summary linked to this article.

## Data availability

The identified dimeric structures of CCR5, CXCR4, and CXCR4–CCR5 are available as pdb files in the Source Data file and in the Zenodo database under accession code https://doi.org/10.5281/zenodo.8337056. All other data generated in this study and source data for each main and Supplementary Figure, including input files and force

field parameters, have been deposited in the Zenodo database under accession code https://doi.org/10.5281/zenodo.8337056. Source data for each main and Supplementary Figure are also provided in this paper.

## Code availability

The CG–MetaD protocol employed in this work is available on PLUMED-NEST under project id plumID:23.014 [https://www.plumed-nest.org/eggs/23/014/]. Python code for evaluation of the lowest energy path (LEP) is available on GitHub. Backmapping scripts from CG to atomistic structures and all PLUMED input files are included in the Zenodo database under accession code https://doi.org/10.5281/zenodo.8337056. Other analysis tools written in R or Python used in the current study are available from the corresponding author on request.

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

## Acknowledgements

This work has received funding from the European Research Council (ERC) under the European Union's Horizon 2020 research and innovation program ("CoMMBi" ERC grant agreement No.101001784) and it was supported by a grant from the Swiss National Supercomputing Centre (CSCS) under project ID s1150, and from "Partnership for Advanced Computing in Europe" (PRACE) (call 16 with project ID 2016153685). We also thank the NVIDIA Corporation for the donation of a Tesla K40 GPU. D.D.M. acknowledges the support of the Italian Foundation for Cancer Research AIRC (Project No. IG 2022 ID 27534). The authors thank Stefano Raniolo for useful discussions.

## Author contributions

All authors contributed equally to this work. V.L. devised and supervised the project, V.L., S.M., D.D.M., and P.C. designed the simulations, and S.M., D.D.M., and P.C. performed the calculations and the analysis. P.C. prepared the supplementary movie. All authors analyzed and discussed the results and implications, wrote the main paper and the Supplementary Information, and commented on the paper at all stages.

## Competing interests

The authors declare no competing interests.
