## [Peer Review File · Nature Communications]

Structural basis of dimerization of chemokine receptors CCR5 and CXCR4Reviewers' comments:

Reviewer #1 (Remarks to the Author):

In this paper, the authors predicted the dimer structures of two GPCRs in biological membrane using computer simulations. They examined CCR5 homodimer, CXCR4 homodimer, CCR5 and CXCR4 heterodimer structures using coarse-grained metadynamics simulations. In their metadynamics simulations, they used relative orientation (torsion angle) and distance as collective variables and tried to explore wide conformational spaces of the above three systems. Use of coarse-grained models (MARTINI) as well as an enhanced sampling method (MTD) accelerated their MD simulations significantly and they obtained bound states, pre-bound, and unbound states for the three systems. Their predicted structures include symmetric dimers as well as asymmetric dimers for the three systems, suggesting that dynamic equilibrium may exist between multiple dimer structures in biological membranes. They also suggested that cholesterol may bind at the interface of two GPCRs, which increases the stability of the dimer structures.

In my opinion, the paper contains three major issues as discussed below:

(1) comparison between X-ray structures and major predicted dimers.

Although they compared X-ray and simulated structures in Figure 5D, I don't know how they selected from the simulated structures. If I compare Figure 1 with Figure 5, the structures are very different although interfacial trans-membrane helices are common. In 3OE9, top of TM helices are contact with each other, while in Figure 1, bottom helices are aligned. For this reviewer, they are totally different structures.

First, the authors should have marks of the major and X-ray structures in their landscapes (Figure 5A, 5B, and 5C). Then, my questions will be resolved. If the X-ray and simulate structures are very different, the authors must explain the reasons more clearly. Different membrane environments may change the stability of dimer structures. But, other likely reasons might be insufficient accuracy of the molecular models of proteins and lipid molecules used in the simulations. Did they start the simulations from one of the dimer X-ray structures?

If they did not, I suggest them to carry out such simulations as tests. I believe that clear explanations are necessary if they discuss disagreement between X-ray and simulated structures.

(2) disagreement between torsion angles and shown structures in Figure 5B.

In top left and top middle structures of Figure 5B, their torsion angles are almost the same with each other. However, the dimer interfaces are apparently different. I suspect that it is due to a mistake. Please check the data once again and correct them. I don't know the validity of other structures in

Figure 5. One way to reduce such mistakes would be to add information about torsion angle and distance of each structure in Figure 5A, 5B, and 5C.

(3) binding mechanisms from the simulations.

They discussed unbound, pre-bound, and bound structures in the maintext. However, their relationships and mechanisms for how major dimer structures are formed are missing in the current manuscript. For this, they need to reweight the simulation data and draw free-energy landscapes with different CVs from metadynamics simulations. It requires additional efforts, while to understand the binding mechanisms, I think that it is important.

Reviewer #2 (Remarks to the Author):

The work from Di Marino et al. uses coarse-grained metadynamics to investigate the dimerization of the chemokine receptors CXCR4, CCR5 and the heterodimer CXCR4/CCR5.

The work provides the first insights into the long term dynamics of these receptors in response to an agonist stimulation. Notably, the observation of ligand induced dimerization, and of two stable dimeric states (A symmetric, and B asymmetric) is observed in the authors simulations, in line with previous experimental results (e.g. Isbilir et al. 2020). Furthermore, dimerization interfaces for the CXCR4 homodimers are also in agreement with experimental data, in particular highlighting the role of residues K239, V242 and L246 in determining the asymmetric dimer structure.

The work is well written and the figures are clear, using colorscale to identify the transmembrane domains.

As a note, the use of pink (TM5) and orange (TM6) could be confusing, and a separate color is advised. Furthermore, the axes in the insets in Figure 1 are non legible.

The corresponding author should use, where possible, an institutional affiliation and not is gmail account.

Reviewer #3 (Remarks to the Author):

Di Marino et al. report results of a coarse-grained simulation study of the homo- and heterodimerization of the chemokine receptors CCR5 and CXCR4. This addresses an important question since these dimers form potentially very interesting pharmaceutical targets. Dimer structures obtained with the help of a timely metadynamics approach display both symmetric and asymmetric configurations. The discussed interfaces appear reasonable and consistent with experiments where comparison is possible.

The authors are overall not particularly sparing with words about the importance of such dimer structures for drug design and allosteric modulation. Yet, the work is limited to the *in silico* study of dimerization only. This is a bit disappointing. Additionally, the dimerization of these chemokine receptors has been studied before in quite some detail, yielding as far as I can see similar structures and binding energies. Only the chosen approach here is different. Still, these previous studies are hardly discussed and poorly referenced here. Pinpointing the differences in the results would provide valuable insight into strengths and limitations of the different approaches.

The authors claim to have studied the receptor dimerization in a plasma membrane. The wording 'plasma membrane' for a symmetric lipid bilayer containing only POPC and 10% cholesterol is neither timely nor does it meet the requirements. It has e.g. already been shown that multiply unsaturated lipids like found within the cytosolic membrane leaflet are particularly prone to adsorb to the GPCR surface. How would this change receptor dimerization?

Reviewer #4 (Remarks to the Author):

The manuscript by Di Marino et al investigates the dimerization and heterodimerization of the chemokine receptors CCR5 and CXCR4 that are important members of the G protein-coupled receptor family, by molecular dynamics. These two receptors play a key role for the regulation of vital processes such as leukocyte migration, and they are associated to cancer and neurodegenerative diseases. The method used is a free-energy technique named Coarse-Grained MetaDynamics. Understanding how the dimerization/heterodimerization controls the activity of GPCRs *in vivo* is important. But investigating the molecular basis of these processes are technically challenging. Therefore, molecular dynamics simulation is an interesting and useful approach. The group is expert in molecular simulations of membranes proteins. Here the authors performed long timescale molecular simulations, equivalent to the minute timescale ("5.5 milliseconds of enhanced sampling simulations – corresponding to minute real time"). The results showed that CXCR4 and CCR5 form both symmetric and asymmetric dimers when they are investigated either as homodimers or heterodimers. The transmembrane helices TM4-TM5 and TM6-TM7 are the preferred binding interfaces in these different dimers. Overall the study is mostly descriptive. The main results describe the interface of dimers. In order to bring new information in the field of the molecular bases of GPCR dimerization, it would be of interest to perform molecular

simulations in the presence of ligands (agonist/antagonist), G protein and different compositions of lipids.

My enthusiasm is strongly dampened by the major comments below.

Major comments:

1- The study suffers strongly from the absence of validation of the results by experimental experiments either with reconstituted purified proteins, or in a cellular context. The authors only refer to experimental results from the literature, but there is no tentative to reproduce some of them in their molecular simulations.

2- Figure 4: I'm not convinced by the data provided in this figure, especially the access to the G protein binding site. The variation of distances between TM5 and TM6 (Fig. 4D-F) are very low (only 1-2 angstroms).

3- For a better understanding of the molecular basis of dimerization, the authors should try to disrupt the dimerization/heterodimerization by introducing mutations in the transmembrane helices, and repeat the molecular simulation with these mutants. As stated by the authors (p. 8), TM6 mutations that attenuate CXCR4 dimerization has been reported (Isbilir et al (2020) PNAS). In addition, mutations that destabilize GPCR homodimers have been characterized for other receptors (i.e. in rhodopsin, see Ploier et al, Nat Comm, 2016).

4- How the structural dynamics of the receptors that is controlled by the ligands (agonists or antagonists) has an impact on their homo/heterodimerization? Indeed, in a recent study by Moller et al (Sci Reports, 2018), both antagonist and agonist have been shown to favor oligomerization of the mGlu2 receptor, most probably by limiting its structural dynamics.

5- In addition, the G protein controls the structural dynamics of the GPCRs. How the G protein is expected to influence the homo/heterodimerization of CXCR4 and CCR5 in the method used here? Indeed, the structure of the complex CCR5-Gi was recently reported (Isaikina et al (2021) Sci Adv). The structure of a similar complex was also reported for the chemokine receptor CXCR2 (Liu et al (2021) Nature). Does a mini-G protein could be added in the molecular simulations?

6- The lipids in the membrane (i.e. cholesterol) and lipid modifications of the receptor (i.e. palmitoylation) were reported to influence dimerization of GPCRs. The molecular simulations were performed in membranes composed by POPC and cholesterol molecules in a 9:1 ratio. Do higher concentrations of cholesterol (30% or higher) would influence the dimerization (stability of dimers, dimer interfaces).

Minor comments:

- page 2: “close-to-physiological” and “virtual microscope” are not adapted. Could the authors remove this metaphor, and these two sentences?
- Fig. 4, panel H: “intecellular” should be replaced by “intracellular”.
- Discussion, First paragraph (p13): a long repetition of the results. It should be avoided.

Reviewer 1

1) Comparison between X-ray structures and major predicted dimers.

Although they compared X-ray and simulated structures in Figure 5D, I don't know how they selected from the simulated structure. First, the authors should have marks of the major and X-ray structures in their landscapes (Figure 5A, 5B, and 5C). Then, my questions will be resolved. Did they start the simulations from one of the dimer X-ray structures? If they did not, I suggest them to carry out such simulations as tests. I believe that clear explanations are necessary if they discuss disagreement between X-ray and simulated structures.

We thank the Reviewer for raising this point and we agree that a more detailed discussion on the X-ray dimer structures of CCR5 and CXCR4 is helpful for a better understanding of the power of our methodology and the impact of our results. Following the suggestion of the Reviewer, the X-ray dimer structures of CCR5 and CXCR4 are marked by a white cross in the FES of the revised Figure 5. As shown, they represent energy metastable states at higher inter-promoter distance (CV1) if compared with the lowest energy dimer states. This is due to the presence of bulky crystallization adjuvant molecules - rubredoxin and lysozyme for CCR5 and CXCR4, respectively - which impede by steric hindrance a closer contact between receptors (see Figure C1 below).

We remark that our methodology allows sampling the whole free-energy landscape, visiting and energetically evaluating several times all the possible dimer and monomer states, regardless of the starting structure, which is, in our case, the protomers in the unbound (monomer) state, as

reported at page 3 of the original manuscript. Among the energetically evaluated states are also the X-ray structures of CCR5 (PDB ID 4MBS) and CXCR4 (PDB IDs 3OE9, 3OE8, 3ODU), which are visited a number of times during our simulations as shown in Figure C2 below.

In addition, prompted by the Reviewer's suggestion, we performed 160 μ s of plain coarse-grained molecular dynamics calculations on the X-ray structures of CCR5 and CXCR4 without the crystallization adjuvant molecule. The results show that the X-ray starting pose is left in favour of the closer, energetically more stable dimer structures. Interestingly, in CXCR4 the simulation lands in the lowest energy dimer structure A since no significantly high energy barrier separates the starting X-ray structure from the lowest energy one (see Figure C3 below). We further note that during 160 μ s of plain simulations, the system only partially explores the free energy landscape. This is due to the slow receptor diffusion in membrane, the many energy basins to visit and barriers to cross, which make necessary the employment of enhanced sampling techniques like coarse-grained metadynamics for the full exploration of the free energy landscape.

We have acknowledged the Reviewer's comment adding a more detailed discussion on the X-ray structures on page 12 of the revised manuscript and revising Figure 5, while the simulations results of the X-ray structures are reported in the Supplementary Figure S10.

Fig C1: Representation of the X-ray structures of CXCR4 and CCR5. The receptors dimers of CXCR4 and CCR5 are displayed as coloured cartoons, while the bulky crystallization adjuvant molecules (lysozyme and rubredoxin) as grey cartoon.

Fig C2: Exploration of the distance CV during the CG-MetaD calculations of the CCR5 and CXCR4 homodimers. *Left*) The red dots highlight when the CCR5 homodimer X-ray structure (PDB ID 4MBS) is sampled. *Right*) The green and light blue dots highlight when the CXCR4 homodimer X-ray structures (PDB IDs 3OE9 and 3OE8, respectively) are sampled. The red dashed lines indicate the boundary between simulation walker.

Fig C3: Exploration of the dimerization free energy surface (FES) of CXCR4 during plain CG-MD simulations starting from the X-ray dimer structure PDB ID 3OE9. The region of the FES sampled during 160 μ s of CG-MD simulations is shown as dashed white line. The white X represents the starting X-ray structure PDB ID 3OE9, whereas the red X indicates the final state reached by the system. On the background, the dimerization FES of CXCR4 obtained from CG-MetaD calculations is represented.

2) *Disagreement between torsion angles and shown structures in Figure 5B. In top left and top middle structures of Figure 5B, their torsion angles are almost the same with each other. However, the dimer interfaces are apparently different. I suspect that it is due to a mistake. Please check the data once again and correct them. One way to reduce such mistakes would be to add information about torsion angle and distance of each structure in Figure 5A, 5B, and 5C.*

We thank the Reviewer for raising this point and we do apologise for the mistake. The correct structures are now reported in the revised Figure 5. In addition, the distance and torsion angle CV values for each structure in Figure 5 are reported as suggested by the Reviewer. Thank you!

3) *Binding mechanisms from the simulations. They discussed unbound, pre-bound, and bound structures in the maintext. However, their relationships and mechanisms for how major dimer structures are formed are missing in the current manuscript. It requires additional efforts, while to understand the binding mechanisms, I think that it is important.*

Prompted by the Reviewer’s suggestion, we have characterised the dimerization process calculating the Lowest Energy Paths (LEPs) connecting the monomer (unbound) state to the dimer (bound) states in the three systems, CCR5-CCR5, CXCR4-CXCR4 and CCR5-CXCR4. The results are shown in Figure C4 below. In detail, for each system LEPs were calculated by subdividing the dimerization free energy surface into a grid and, starting from one of the energy minima and metastable states discussed in the main text, we have minimized the free energy of each forward step in the grid. This is done by increasing values of protein-protein distance with respect to the previous one. Minimization was performed in an iterative way, by repeating the calculation several times using different starting states and grid sampling parameters until reaching a converged result. This analysis showed that dimerization is a multistep process passing through one or more metastable states before reaching the lowest energy dimer structures.

The discussion of the LEPs and the dimerization mechanism, with a description of the energetically relevant metastable states, has been added to the “*Dimerization Mechanism*” section at page 12 of the revised manuscript. The images of the computed LEPs shown below have been also added to the revised Figure 5 of the main text.

Fig C4: CCR5 and CXCR4 dimerization mechanism. Representation of the dimerization FES and the structures of the lowest energy minima and metastable states for CCR5-CCR5 (A), CXCR4-CXCR4 (B) and CCR5-CXCR4 (C) systems. The lowest energy structures for each system are indicated as A and B, whereas the metastable states are depicted as Greek letters. The values of distance and torsion CVs (d and t) are reported under each structure. White crosses show the position on the FES of the X-ray CCR5 and CXCR4 dimer structures. The Lowest Energy Paths (LEP) connecting the monomer unbound state to the energy dimer minima are shown as black, red, or purple solid lines. The transitions between energy minima and metastable states identified by the LEPs are represented by arrows connecting the corresponding structures displayed as cartoon above and below the FES.

Reviewer 2

1) The work provides the first insights into the long term dynamics of these receptors...the use of pink (TM5) and orange (TM6) could be confusing, and a separate color is advised. Furthermore, the axes in the insets in Figure 1 are non legible.

We thank the Reviewer for his/her positive comments. We have acknowledged the Reviewer's suggestion changing TM6 color from orange to magenta in all the figures of the manuscript and increasing the font size of the axes labels in Figure 1.

2) The corresponding author should use, where possible, an institutional affiliation and not is gmail account.

Following the Reviewer's request, vittorio.limongelli@usi.ch now replaces vittoriolimongelli@gmail.com.

Reviewer 3

1) Previous studies are hardly discussed and poorly referenced here. Pinpointing the differences in the results would provide valuable insight into strengths and limitations of the different approaches.

We thank the Reviewer for this comment and we agree that a more detailed discussion of the previous works is useful. In the original manuscript, to the best of our knowledge all the previous works relevant to the topic were cited and discussed taking into account the article size limit of the journal. In the revised manuscript, we have acknowledged the Reviewer's request extending the discussion of our results with respect to the state of the art at page 18. For the Reviewer's convenience, the added text is also reported below:

“Previous computational studies proposed CCR5 and CXCR4 dimer structures^{51,52} that are worth comparing with those identified in our work. In particular, Pluhackova et al. showed that CXCR4 can form asymmetric dimers through TM1/TM5-7, whereas cholesterol might stabilize the symmetric TM3-TM4 dimer interface.⁵¹ While such dimer states are found in our study, they turn out to be at higher energy values compared to the symmetric TM1-TM6-TM7 and the asymmetric TM6-TM7/TM5-TM5 dimer interface of structure A and B, respectively, found in our study. In a second work, the same authors investigated CCR5 and CCR5-CXCR4 dimerization.⁵² Here, the most populated dimer interfaces are asymmetric where CCR5 forms dimers through TM1-H8/TM4-TM5 and TM1-H8/TM5-TM6-TM7, whereas CCR5 and CXCR4 interact through TM1 and TM4-TM5-TM6-TM7. At variance with these results, we show that the lowest energy CCR5 and CCR5-CXCR4 dimers have symmetric binding interface composed of TM4-TM5, however low energy asymmetric dimers are also possible through TM1-TM2-H8/TM4-TM5 and TM1-TM2-H8/TM6-TM7 for CCR5 and CCR5-CXCR4 dimers, respectively. The different results might be

attributed to the diverse simulation technique employed. In fact, both in ref 51 and 52 the authors performed standard coarse-grained molecular dynamics simulations, which allow only a partial exploration of the dimerization free energy surface as also shown by our simulations reported in the Supplementary Materials (see Supplementary Figure S10). Differently, here we have combined metadynamics with coarse-grained molecular dynamics. This allows enhancing the sampling of the phase space and achieving a converged dimerization free-energy landscape. In so doing, each possible dimer state is visited and energetically evaluated - including those proposed in ref. 51 and 52 - providing a thorough description of receptor dimerization mechanism. Proof of that is further given by the fact that the X-ray dimer structures of CCR5 and CXCR4 are also identified in our study (see Fig. 5), while they are not found by standard CG-MD simulations reported in refs 51 and 52.”

We did our best to keep the literature updated - and we believe we did - however, in case any contribution relevant to the topic was forgotten, we kindly ask the Reviewer to inform us.

2) The authors claim to have studied the receptor dimerization in a plasma membrane. The wording ‘plasma membrane’ for a symmetric lipid bilayer containing only POPC and 10% cholesterol is neither timely nor does it meet the requirements. It has e.g. already been shown that multiply unsaturated lipids like found within the cytosolic membrane leaflet are particularly prone to adsorb to the GPCR surface. How would this change receptor dimerization?

We have acknowledged the Reviewer’s comment referring to the POPC/cholesterol bilayer as “membrane model” and not as “plasma membrane model” throughout the text of the revised manuscript. In addition, prompted by the Reviewer’s comment, we set up a realistic plasma membrane model, which is asymmetrically composed by differently saturated phospholipids and higher cholesterol concentration, mimicking the composition of in vivo cell membrane (for details see Song *et al. Structure* 2019 doi: 10.1016/j.str.2018.10.024). The list of the components of the plasma membrane model is reported in Table C1 below.

Lipid Family	Molecule	% Outer Leaflet	% Inner Leaflet
PC	POPC	20	5
	DOPC	20	5
PE	POPE	5	20
	DOPE	5	20
Sph	POSM	15	

GM3	DPG3	10	
Chol	Chol	25	25
PS	POPS		8
	DOPS		7
PIP2	DPP2		10

Table C1: Composition of the in vivo mimicking membrane model. Lipid family, molecule and concentration (percentage) in the outer and inner leaflet is reported. Table legend: PC = phosphatidylcholines; POPC = 1-palmitoyl-2-oleoyl-sn-glycero-3-phosphatidylcholines; DOPC = 1,2-dioleoyl-sn-glycero-3-phosphatidylcholines; PE = phosphatidylethanolamine; POPE = 1-palmitoyl-2-oleoyl-sn-glycero-3-phosphoethanolamine; DOPE = 1,2-dioleoyl-sn-glycero-3-phosphoethanolamine; Sph = sphingolipid; POSM = N-(9Z-octadecenoyl)-hexadecasphing-4-enine-1-phosphocholine; GM3 = monosialodihexosylganglioside; DPG3 = neuAcalpha2-3Galbeta1-4Glcbeta-Cer(d16:1/16:0); Chol = cholesterol; PS = phosphatidylserine; POPS = 1-hexadecanoyl-2-(9Z-octadecenoyl)-sn-glycero-3-phosphoserine; DOPS = 1,2-di-(9Z-octadecenoyl)-sn-glycero-3-phosphoserine; PIP2 = phosphatidylinositol 4,5-bisphosphate; DPP2 = CG model corresponding to the atomistic C16:0 dipalmitoyl phosphatidylinositol 4,5-bisphosphate (DP-PIP2) – C18:0 distearoyl phosphatidylinositol 4,5-bisphosphate (DS- PIP2).

In order to study the CCR5 and CXCR4 dimer structures in the plasma membrane model, we performed additional 300 μ s coarse-grained molecular dynamics calculations. The results show that the dimer structures identified in the original manuscript are also found in the plasma membrane model (see Figure C5 below). However, in CCR5 and CCR5-CXCR4 dimer A the minimum CV values are slightly increased. This is due to the presence of a DOPE and cholesterol molecule at the dimer interface that mediate the binding interaction between protomers, without changing the original binding interface. Our results in the plasma membrane systems demonstrate that POPC is a reliable, “simple” membrane model to identify receptor dimers, however certain lipids might interplay with protomers during dimerization. Further investigations are planned in the near future to characterize more deeply the GPCR/lipids interaction in plasma membrane models during receptor dimerization and oligomerization.

The results of the additional calculations in the plasma membrane model are reported in a new section entitled “*Effect of membrane lipid composition and G Protein coupling on receptor dimerization*” of the revised manuscript at pages 14-15.

Fig C5: CCR5 and CXCR4 dimers in plasma membrane model. (A) Composition of the plasma membrane model with detail of the outer (top) and inner (bottom) leaflets and representation of the proteins in membrane (center). The proteins are shown as silver surfaces, whereas the lipids are colored in lime (POPC), dark green (POPS), green-yellow (POPE), turquoise (cholesterol), medium violet red (DOPC), light coral (DOPS), pink (DOPE), cyan (DPP2), green (POSM), and chartreuse (DPG3). The hydrogen atoms of the lipids are represented as white sphere, whereas oxygens are depicted in red and nitrogens in blue. On the center, exploration of the FES of the lowest energy dimer structures of CCR5 (A), CXCR4 (B) and CCR5-CXCR4 (C) embedded in the plasma membrane model. The white dashed line represents the region of the FES sampled by each system during 50 μ s of CG-MD simulations. The starting points of each calculation is highlighted by white crosses. On the background, the dimerization FES for each system computed with CG-MetaD calculations in the POPC/cholesterol 9:1 bilayer is shown. On the right, the insets show the comparison between the quaternary structures of CCR5 (B) and CCR5-CXCR4 (D) dimer A in the plasma membrane model (depicted in color-coded cartoons) and in the POPC/cholesterol membrane model (gray cartoons). DOPE and cholesterol molecules interposed between the protomers in the dimers obtained in the plasma membrane models are represented as spheres with carbons coloured in purple, nitrogens in blue, oxygens in red, and sulfur atoms in orange.

Reviewer 4

1) The study suffers strongly from the absence of validation of the results by experimental experiments either with reconstituted purified proteins, or in a cellular context. The authors only refer to experimental results from the literature, but there is no tentative to reproduce some of them in their molecular simulations.

We agree with the Reviewer - and we are well aware - that computational results need experimental validation and in our case our results are validated by the available mutagenesis data on CCR5 and CXCR4. It is important noting that our simulation parameters do not contain any information regarding CCR5 and CXCR4 mutations known to affect receptor dimerization. These are identified *a posteriori* by analysing our results. Prompted by the Reviewer's comment, we have performed additional simulations on mutant forms of CCR5 and CXCR4 affecting

dimerization that confirm our findings. These results are discussed at point 3 (see below) and in the revised manuscript at page 17.

Finally, we note that we are a theoretical group and do not perform experiments. In recent years, computations have reached an unprecedented level of accuracy - comparable to that of experiments - as demonstrated by many works including the renowned AlphaFold2 contribution (*Jumper et al. Nature 2021 doi: 10.1038/s41586-021-03819-2*). Our techniques are recognised milestone binding free energy methods in protein systems (e.g., please see *Fu et al. J Med Chem 2022 doi: 10.1021/acs.jmedchem.2c00796*; *Bussi & Laio Nat Rev Phys 2020 doi: 10.1038/s42254-020-0153-0*) and for proposing them to study GPCRs we have received an ERC grant (CoMMBi ERC grant). We believe that the CCR5 and CXCR4 dimer structures disclosed - and released as PDB files - for the first time in a membrane environment where protomers are fully free to move and orient, and each possible dimer state is energetically evaluated, represent a significant advance in the field. This paves the way to further structure-based investigations on relevant GPCRs functions like inter-receptor allosteric regulation and receptor (not ligand) biased signaling, just to mention a few. These are all intriguing questions raised by our study that we - as GPCR community - are asked to address in the coming years.

2) *Figure 4: I'm not convinced by the data provided in this figure, especially the access to the G protein binding site. The variation of distances between TM5 and TM6 (Fig. 4D-F) are very low (only 1-2 angstroms).*

We thank the Reviewer for raising this point. We agree that the original Figure 4D-F was not clear, therefore we have replaced it with the new Figure 4D-F. Here, we plot the distance distribution between transmembrane 6 (residue Arg232^{6.32} for CCR5 and Lys236^{6.32} for CXCR4) and transmembrane 3 (residue Arg126^{3.50} for CCR5 and Arg134^{3.50} for CXCR4) that is a widely used hallmark of the receptor conformational change allowing the G protein binding and in turn its activation. A similar distance has been also used to define A2AGPCR activation (*Carpenter et al. Nature 2016 doi: 10.1038/nature18966*). The new figure reports as reference values the same distance distribution computed in the experimental inactive and active structures of CCR5 and in the inactive structures of CXCR4, since the active state of CXCR4 has not been resolved yet. The plots confirm what was reported in the original manuscript. In particular, the CCR5 homodimer structure A induces a conformational change towards the active form of one of the two protomers (Figure 4A), while both the CCR5-CXCR4 heterodimer structures shift CXCR4 towards the inactive state. This evidence indicates that dimerization represents *de facto* an allosteric regulatory mechanism of GPCR activity.

The new Figure 4 and its discussion is reported at pages 10-11 of the revised manuscript.

3) For a better understanding of the molecular basis of dimerization, the authors should try to disrupt the dimerization/heterodimerization by introducing mutations in the transmembrane helices, and repeat the molecular simulation with these mutants. As stated by the authors (p. 8), TM6 mutations that attenuate CXCR4 dimerization has been reported (Isbilir et al (2020) PNAS).

We have acknowledged the Reviewer's request performing additional calculations on all the CCR5 and CXCR4 mutants that are reported in literature to affect receptor dimerization (Isbilir et al. PNAS 2020; Martínez-Muñoz et al. Molecular Cell 2018; Colin et al. PLoS Pathog. 2018; Hernanz-Falcón et al. Nat. Immunol. 2004; Jin et al. Sci. Signal. 2018). These mutations are listed in Supplementary Table 4. In detail, we employed two different algorithms to estimate the effect for each mutation on the protomer-protomer binding energy for the dimer structures identified in our study. In addition, we performed the same calculations for the X-ray dimer structures. The results are reported in Supplementary Table 4 and in Table C2 below.

System	Minimum	Protomer	Mutations	$\Delta\Delta G$	
				MutaBind2	mCSC-PPI2
CCR5 - CCR5	A	A	L196 ^{5.40} K	-0.84	-0.25
			I200 ^{5.44} K	-0.89	-0.689
			L205 ^{5.49} K	-1.26	-0.786
	B	B	V150 ^{4.47} A	-0.89	0.187
			L196 ^{5.40} K	-0.91	-0.26
			I200 ^{5.44} K	-0.94	-0.473
	B	A	L205 ^{5.49} K	-0.9	-0.231
L196 ^{5.40} K			-0.82	-0.708	
B	B	I200 ^{5.44} K	-1.39	-0.811	
		L205 ^{5.49} K	-1.08	-0.656	
4MBS	B	I52 ^{1.54} V	-0.22	-0.182	
		V150A	-0.08	-0.09	
CXCR4 - CXCR4	A	A	K239 ^{6.35} E	-0.16	-0.964
			L246 ^{6.42} A	-0.73	-0.14
			L246 ^{6.42} P	-1.48	-0.748
	B	A	K239 ^{6.35} E	-0.35	-0.772
			K239 ^{6.35} E	-1.68	-0.961
			V242 ^{6.38} D	-0.97	-0.067
			V242 ^{6.38} A	-0.56	0.201
			L246 ^{6.42} A	-2.08	-1.217
	3OE8	B	L246 ^{6.42} P	-2	-1.274
			K239 ^{6.35} E	-0.2	-0.289
3OE8	B	V242 ^{6.38} D	-0.54	-0.061	
		V242 ^{6.38} A	-0.33	-0.123	
3OE9	N/A	N/A	N/A	N/A	

Table C2: Effects of mutations on CCR5 and CXCR4 dimer structures. The effect of each mutation is assessed by computing the change in protomer-protein binding energy using MutaBind2 and mCSC-PPI2 algorithms.

As can be seen, most of the mutations significantly affect the energetic stability of the CCR5 and CXCR4 dimer structures identified in our study, while they do not play a role in the X-ray ones. Among these, L196K^{5.40}, I200K^{5.44} and L205K^{5.49} are disruptive on both the dimer structures A and B of CCR5, while K239E^{6.35} and L246P^{6.42} affect both the dimer structures A and B of

CXCR4. The discussion of the effect of CCR5 and CXCR4 mutations on dimerization is provided at page 17 of the revised manuscript.

Furthermore, in order to identify novel mutations affecting receptor dimerization, we have mutated into alanine (Ala scan) each residue closer than 8 Å to the binding interface in the CCR5 and CXCR4 dimer structures, and reported the most disruptive ones - with binding energy alteration higher than 1 kcal/mol - in Supporting Table 5. Our in silico mutagenesis experiments show that three tryptophan residues involved at the binding interface of both homo- and heterodimers- i.e., Trp190^{5.34} (CCR5), Trp283^{7.34} and Trp195^{5.34} (CXCR4) - significantly contribute to the energetic stability of the dimer structures. Our results prompt to further investigate their role in CCR5 and CXCR4 dimerization using molecular simulations possibly in combination with spectroscopic experiments such as those employing 5-¹³C-methyl-deutero- tryptophan TROSY-NMR technology, capable of detecting conformational changes of tryptophan residues during GPCR dimer formation.

4) How the structural dynamics of the receptors that is controlled by the ligands (agonists or antagonists) has an impact on their homo/heterodimerization? Indeed, in a recent study by Moller et al (Sci Reports, 2018), both antagonist and agonist have been shown to favor oligomerization of the mGlu2 receptor, most probably by limiting its structural dynamics.

We thank the Reviewer for raising this point and we agree that ligand binding might play a role in GPCR dimerization. Providing accurate structural details of the effect of ligand binding on receptor dimerization is challenging since the dynamics of both ligand binding and receptors motion should be concurrently investigated. This requires taking into account a large number of degrees of freedom and even longer timescales that are not accessible using the state of the art techniques. We are working to include ligand binding effect on GPCR dimerization in the updated version of our protocol since it also represents a major objective of our ERC project.

In the meantime, in order to accommodate the Reviewer's request, we have performed additional coarse-grained molecular dynamics calculations on CCR5 in the active state - i.e., in the presence of the agonist Chemokine C-C Motif Ligand 3 (CCL3) bound to the orthosteric binding site and the G protein coupled to the intracellular binding site, henceforth defined as aCCR5(G). In particular, we performed 200 μs calculations on CCR5 homodimer and CCR5-CXCR4 heterodimer. Our results show that in both systems the dimer structure A is preserved with the protomers slightly more distant due to the presence of the G protein (see Fig. C6 below). On the other hand, the dimer structures B change. In fact, the binding interface TM1-TM2-TM8 is not used by the active CCR5 (aCCR5(G)), which instead binds to the other protomer - CCR5 or CXCR4 - always through TM4-TM5, suggesting a G protein-mediated selection mechanism of dimerization interface. Such finding further endorses dimerization as a fine regulatory mechanism of receptor activity - alternative to ligand biased signaling - in which a certain dimer

state assumed by the receptor might favour or disfavour the binding of G protein, beta-arrestin or another effector protein, and in turn triggers a specific signal cascade.

We have discussed the results of the additional calculations in a new section of the revised manuscript called “*Effect of membrane lipid composition and G Protein coupling on receptor dimerization*”.

Fig C6: Effect of CCR5 agonist and G protein on CCR5 and CCR5-CXCR4 dimers. Exploration of the FES of the aCCR5(G)-CCR5 (A) and aCCR5(G)-CXCR4 (D) dimers in the complex bilayer, where aCCR5(G) is the activated CCR5 protein bound to CCL3 and the $G\alpha\beta\delta$ heterotrimer. The white dashed line represents the CV values sampled during the CG-MD simulations. The backmapped atomistic structure representing the most populated clusters of A^G and B^G minima of aCCR5(G)-CCR5 are reported on the right as color coded cartoon in panel B and C, respectively. The CCR5 dimer structures A and B are reported on the left. Similarly, The backmapped atomistic structure representing the most populated clusters of A^G and B^G minima of aCCR5(G)-CXCR4 are reported on the right as color coded cartoon in panel E and F, respectively. The CCR5-CXCR4 dimer structures A and B are reported on the left.

5) In addition, the G protein controls the structural dynamics of the GPCRs. How the G protein is expected to influence the homo/heterodimerization of CXCR4 and CCR5 in the method used here? Indeed, the structure of the complex CCR5-Gi was recently reported (Isaikina et al (2021) Sci Adv). The structure of a similar complex was also reported for the chemokine receptor CXCR2 (Liu et al (2021) Nature). Does a mini-G protein could be added in the molecular simulations?

We have acknowledged the Reviewer's request simulating the dimerization of the full active state of CCR5 bound to the G protein trimer - Gαβδ heterotrimer - and the CCL3 agonist ligand, as reported in the PDB structure 7F1Q. We limit our study to CCR5 as no experimental structure of the full active state of CXCR4 coupled with G protein is available so far. Please see previous point 4 for discussion.

6) The lipids in the membrane (i.e. cholesterol) and lipid modifications of the receptor (i.e. palmitoylation) were reported to influence dimerization of GPCRs. The molecular simulations were performed in membranes composed by POPC and cholesterol molecules in a 9:1 ratio. Do higher concentrations of cholesterol (30% or higher) would influence the dimerization (stability of dimers, dimer interfaces).

We agree with the Reviewer that different types of phospholipids and cholesterol concentration do play a role in receptor dimerization. Prompted by her/his comment, we set up a realistic plasma membrane model, which is asymmetrically composed by differently saturated phospholipids and higher cholesterol concentration, mimicking the composition of in vivo cell membrane (for details see Song et al. Structure 2019 doi: 10.1016/j.str.2018.10.024). The list of the components of the plasma membrane model is reported in Table C3 below.

Lipid Family	Molecule	% Outer Leaflet	% Inner Leaflet
PC	POPC	20	5
	DOPC	20	5
PE	POPE	5	20
	DOPE	5	20
Sph	POSM	15	
GM3	DPG3	10	

Chol	Chol	25	25
PS	POPS		8
	DOPS		7
PIP2	DPP2		10

Table C3: Composition of the in vivo mimicking membrane model. Lipid family, molecule and concentration (percentage) in the outer and inner leaflet is reported. Table legend: PC = phosphatidylcholines; POPC = 1- palmitoyl-2-oleoyl-sn-glycero-3-phosphatidylcholines; DOPC = 1,2-dioleoyl-sn-glycero-3-phosphatidylcholines; PE = phosphatidylethanolamine; POPE = 1-palmitoyl-2-oleoyl-sn-glycero-3-phosphoethanolamine; DOPE = 1,2-dioleoyl-sn-glycero-3-phosphoethanolamine; Sph = sphingolipid; POSM = N-(9Z-octadecenoyl)-hexadecasphing-4-enine-1-phosphocholine; GM3 = monosialodihexosylganglioside; DPG3 = neuAcalpha2-3Galbeta1-4Glcbeta-Cer(d16:1/16:0); Chol = cholesterol; PS = phosphatidylserine; POPS = 1-hexadecanoyl-2-(9Z-octadecenoyl)-sn-glycero-3-phosphoserine; DOPS = 1,2-di-(9Z-octadecenoyl)-sn-glycero-3-phosphoserine; PIP2 = phosphatidylinositol 4,5-bisphosphate; DPP2 = CG model corresponding to the atomistic C16:0 dipalmitoyl phosphatidylinositol 4,5-bisphosphate (DP-PIP2) – C18:0 distearoyl phosphatidylinositol 4,5-bisphosphate (DS- PIP2).

In order to study the CCR5 and CXCR4 dimer structures in the plasma membrane model, we performed additional 300 μ s coarse-grained molecular dynamics calculations. The results show that the dimer structures identified in the original manuscript are also found in the plasma membrane model (see Figure C7 below). However, in CCR5 dimer A and CCR5-CXCR4 dimer B the minimum CV values are slightly increased without changing the original binding interface. This is due to the presence of a DOPE and cholesterol molecule at the dimer interface that mediate the binding interaction between protomers. Our results in the plasma membrane systems demonstrate that POPC is a reliable, “simple” membrane model to identify receptor dimers, however certain lipids might interplay with protomers during dimerization. Further investigations are planned in the near future to characterize more deeply the GPCR/lipids interaction in plasma membrane models during receptor dimerization and oligomerization.

The results of the additional calculations in the plasma membrane model are reported in a new section entitled “*Effect of membrane lipid composition and G Protein coupling on receptor dimerization*” of the revised manuscript at pages 14-15.

Fig C5: CCR5 and CXCR4 dimers in plasma membrane model. (A) Composition of the plasma membrane model with detail of the outer (top) and inner (bottom) leaflets and representation of the proteins in membrane (center). The proteins are shown as silver surfaces, whereas the lipids are colored in lime (POPC), dark green (POPS), green-yellow (POPE), turquoise (cholesterol), medium violet red (DOPC), light coral (DOPS), pink (DOPE), cyan (DPP2), green (POSM), and chartreuse (DPG3). The hydrogen atoms of the lipids are represented as white sphere, whereas oxygens are depicted in red and nitrogens in blue. On the center, exploration of the FES of the lowest energy dimer structures of CCR5 (A), CXCR4 (B) and CCR5-CXCR4 (C) embedded in the plasma membrane model. The white dashed line represents the region of the FES sampled by each system during 50 μ s of CG-MD simulations. The starting points of each calculation is highlighted by white crosses. On the background, the dimerization FES for each system computed with CG-MetaD calculations in the POPC/cholesterol 9:1 bilayer is shown. On the right, the insets show the comparison between the quaternary structures of CCR5 (B) and CCR5-CXCR4 (D) dimer A in the plasma membrane model (depicted in color-coded cartoons) and in the POPC/cholesterol membrane model (gray cartoons). DOPE and cholesterol molecules interposed between the protomers in the dimers obtained in the plasma membrane models are represented as spheres with carbons coloured in purple, nitrogens in blue, oxygens in red, and sulfur atoms in orange.

Minor points:

- page 2: “close-to-physiological” and “virtual microscope” are not adapted. Could the authors remove this metaphor, and these two sentences?

Done.

- Fig. 4, panel H: “intecellular” should be replaced by “intracellular”.

Done.

- Discussion, First paragraph (p13): a long repetition of the results. It should be avoided.

Done.

We thank again all the Reviewers for their work and useful suggestions. We believe that the revised manuscript is now ready for publication in *Nature Communications*.

Looking forward to hearing from you.

Sincerely,

Vittorio Limongelli

REVIEWER COMMENTS

Reviewer #1 (Remarks to the Author):

In this revision, the authors replied all the questions and concerns from reviewers sufficiently. The manuscript has become more clear compared to the original one. It is still concern that the results are validated only by limited number of experimental data. In this sense, their mutagenesis simulations are very helpful for readers to understand the reliability of their methods and simulations.

Reviewer #3 (Remarks to the Author):

See attachment

Overall, I liked the study. However, after taking a closer look at previous studies, disappointment break ground. This work tackles the same question (homo- and heterodimerization of CXCR4 and CXCR5) as previous simulation work by Pluhackova and colleagues (51,52), uses very similar methods, similar system setups (POPC, 10% cholesterol; however, here no comparison for pure POPC and 30% cholesterol), and arrives at similar answers (dimerization as dynamic process, symmetric and asymmetric homo- and heterodimers). Still, even in the revision, the introduction reads as if the authors never heard about previous work on the same subject. Reference to this previous work is now added to the Discussion section, albeit in a rather superficial way (see below).

This work instead excels by application of coarse-grained metadynamics to the above problem, developed by the authors, that should allow for proper estimates of binding free energies. This binding energy was estimated to 21-24 kcal/mol, a value that is 4-8(!) times the binding affinity measured (and computed) between transmembrane helices - and hardly discussed in the manuscript. This is the more disappointing since Pluhackova et al. arrived at more reasonable dimerization free energies of ~20kJ/mol (as lower bound). Here, a reference to Martini 3 would be appropriate, in particular, since Martini 3 was shown to yield results in favorable agreement with experiments (Souza et al. 2021. Nat. Methods. 18:382–388). Martini 3 was seemingly not used here as suggested by the following sentence in the Results and Discussion section, lines 146,147: 'Here, we employed the Martini coarse-grained force field (55–57),..'

It is great to see that the authors added in the revision simulations employing an improved biomembrane model. However, instead of investigating the chemokine receptor dimerization in this biomembrane model (the topic of this study), the authors instead studied the 'stability' of previously identified dimers within this model membrane employing standard Martini simulations. It is well known that Martini transmembrane dimers tend to be glued together, that observing dissociation is highly unlikely even if the dimer would not spontaneously form within this environment. Interestingly, the authors show themselves in Figure S10 that this is not the way to go. It was already shown before that GPCR dimerization may be substantially modified in different lipid environments (probably not yet for chemokine receptors).

Point 1) of initial assessment:

1) Previous studies are hardly discussed and poorly referenced here. Pinpointing the differences in the results would provide valuable insight into strengths and limitations of the different approaches.

We thank the Reviewer for this comment and we agree that a more detailed discussion of the previous works is useful. In the original manuscript, to the best of our knowledge all the previous works relevant to the topic were cited and discussed taking into account the article size limit of the journal. In the revised manuscript, we have acknowledged the Reviewer's request extending the discussion of our results with respect to the state of the art at page 18. For the Reviewer's convenience, the added text is also reported below:

"Previous computational studies proposed CCR5 and CXCR4 dimer structures^{51,52} that are worth comparing with those identified in our work. In particular, Pluhackova et al. showed that CXCR4 can form asymmetric dimers through TM1/TM5-7, whereas cholesterol might stabilize the symmetric TM3-TM4 dimer interface.⁵¹ While such dimer states are found in our study, they turn out to be at

higher energy values compared to the symmetric TM1-TM6-TM7 and the asymmetric TM6-TM7/TM5-TM5 dimer interface of structure A and B, respectively, found in our study."

Reply:

This comparison jumps too short: Diving into the previously published work, both symmetric *and* asymmetric CXCR4 dimers were described previously – similar to this work. The TM1/TM5-7 probably corresponds to the TM1-TM6-TM7 binding interface here. TM6 participation in the interface is thus not a novel observation like suggested by the results description (page 8). Regarding the cholesterol-stabilized interface: A careful comparison shows that the TM3-TM4 was dominant only for 30% cholesterol in Pluhackova et al. At 10% cholesterol, this interface rarely occurred suggesting a lower binding affinity, similar to what is seen here.

In a second work, the same authors investigated CCR5 and CCR5-CXCR4 dimerization.⁵² Here, the most populated dimer interfaces are asymmetric where CCR5 forms dimers through TM1-H8/ TM4-TM5 and TM1-H8/TM5-TM6-TM7, whereas CCR5 and CXCR4 interact through TM1 and TM4-TM5-TM6-TM7. At variance with these results, we show that the lowest energy CCR5 and CCR5-CXCR4 dimers have symmetric binding interface composed of TM4-TM5, however low energy asymmetric dimers are also possible through TM1-TM2-H8/TM4-TM5 and TM1-TM2-H8/ TM6-TM7 for CCR5 and CCR5-CXCR4 dimers, respectively. The different results might be attributed to the diverse simulation technique employed.

Reply:

The authors compare with a dimerization study performed in either 100%POPC or in POPC/30% cholesterol, i.e. a different lipid environment than the one studied here.

For simplified comparison and understanding of the energy landscapes it would be great if the authors could explain in a sketch how the one-dimensional torsion angle can be used to characterize the two-dimensional orientation space for two proteins forming a dimer.

In fact, both in ref 51 and 52 the authors performed standard coarse-grained molecular dynamics simulations, which allow only a partial exploration of the dimerization free energy surface as also shown by our simulations reported in the Supplementary Materials (see Supplementary Figure S10).

Reply:

This is not shown in Figure S10. Figure S10 only compares the limitation of starting from a preformed dimer (from crystal structure). It basically shows that the simulations using a complex lipid composition added in the revision are of little value if one is interested in dimerization. References 51 and 52 instead employ hundreds of simulations starting from different structures with well separated chemokine monomers.

Differently, here we have combined metadynamics with coarse-grained molecular dynamics. This allows enhancing the sampling of the phase space and achieving a converged dimerization free-energy landscape. In so doing, each possible dimer state is visited and energetically evaluated - including those proposed in ref. 51 and 52 - providing a thorough description of receptor dimerization mechanism. Proof of that is further given by the fact that the X-ray dimer structures of CCR5 and CXCR4 are also identified in our study (see Fig. 5), while they are not found by standard CG-MD simulations reported in refs 51 and 52."

Reply:

Again, this assessment of 51,52 is not correct. Rmsd to crystal dimer structures of 3.9A and 4.1A were reported in Pluhackova et al.

Reviewer #4 (Remarks to the Author):

The revised manuscript by Di Marino et al. investigates the dimerization and heterodimerization of the chemokine receptors CCR5 and CXCR4 by molecular dynamics. Using a free-energy technique called Coarse-Grained MetaDynamics, the authors performed long timescale molecular simulations, equivalent to the minute timescale (“5.5 milliseconds of enhanced sampling simulations – corresponding to minute real time”).

The authors have now investigated *in silico* the effect of different lipids and the G proteins, as well as the mutations at the dimer interface. However, I still have major concerns about the study. The first is the lack of validation of the results by experiments.

In addition, the authors state that the receptors studied “prefer forming dimers instead of being in monomeric states” (line 141). And they say that this is in agreement with the results of the Mellado group. This statement is highly misleading because the Mellado group’s results on GPCR dimerization have been heavily debated. In contrast, Lohse’s group provides compelling evidence that CXCR4 receptors form mostly monomers at the cell surface when expressed under physiological (low amount of receptors) conditions (Isbilir et al (2020 PNAS; PMID: 33148803). Unfortunately, this study and reference are not discussed here.

Line 119-120: when the simulations started, the two protomers were placed about 7 nm apart. Could you show a picture of the starting situation in a figure, to better evaluate this distance in relation to the size the protomers?

The quality of the writing should be improved in many parts:

- In most of the figures, plot axis legends are too small to be read.
- Paragraph on “Dimerization mechanism” (line 352-404) is written in too much detail. As a result, it is difficult to get a clear picture.
- The end of the introduction is very long (line 81-104).
- Line 114: delete “in Nature Methods”.
- The G protein should be named “Galphabetagamma” and not “Galphabetadelta”.
- The expression “virtual microscope” (line 69) is not adapted as proposed in my review of the original manuscript.

Reviewer 3

1) *This work instead excels by application of coarse-grained metadynamics to the above problem, developed by the authors, that should allow for proper estimates of binding free energies. This binding energy was estimated to 21-24 kcal/mol, a value that is 4-8(!) times the binding affinity measured (and computed) between transmembrane helices - and hardly discussed in the manuscript. This is the more disappointing since Pluhackova et al. arrived at more reasonable dimerization free energies of ~20kJ/mol (as lower bound).*

We thank the Reviewer for appreciating our methodology. It is important to say that our motivation behind the development of CG-MetaD, originally published in JACS (*doi: 10.1021/jacs.6b05602*), was to alleviate the limitation of enthalpically driven CG force fields like Martini that tend to over-stabilise the bound states. In fact, the metadynamics bias allows the system to escape from one energy minimum and fall in another one, crossing even large energy barriers. Therefore, the concrete advantage of combining Coarse Grained molecular dynamics and MetaDynamics (CG-MetaD) is the enhanced sampling of the phase space. This is also the case of the present study where thanks to CG-MetaD we observed a remarkable number of forth-and-back events between the dimer and monomer states, 566 for CCR5, 146 for CXCR4, and 327 for CCR5-CXCR4, as reported in Table S1 of the original manuscript. These recrossing events ensure a full exploration of the phase space (i.e., all the possible dimer states) leading to the convergence of free-energy calculation (see Fig. S2 of the original manuscript and Fig. 1 below) and *a quantitatively characterised free energy surface*. The latter is necessary to disclose *the low energy - hence most probable - dimer states and the binding mechanism of these receptors* that is

the aim of our study. To this end, *the relative free-energy difference between two or more dimer states is the more important data that allows the identification of the lowest energy dimers*, while the absolute binding free energy value has minor significance since it can be influenced by simulation conditions, including the accuracy of the force field. For this reason, we report the absolute binding free energy estimates in the manuscript without emphasising their biological meaning.

That said, the absolute values of binding free energy reported in our study are not surprising considering that a very similar estimate of binding free energy (~25 kcal/mol) was recently reported for the dimerization of rhodopsin GPCR by Lamprakis et al. (*doi. 10.1021/acs.jctc.0c00507*). These values are far from those (~5 kcal/mol) reported in the Böckmann's studies (*Pluhackova et al. doi. 10.1371/journal.pcbi.1005169* and *Gahbauer, Pluhackova & Böckmann doi. 10.1371/journal.pcbi.1006062*), though all these works employed the same version of the coarse-grained force field (i.e., Martini 2).

At this regard, we note that Böckmann and colleagues refer to their binding free energies as lower bound values (as also noted by the Reviewer), in particular they wrote: "*Lower bound estimates for the binding free energy between chemokine receptors in coarse-grained simulations could be derived from the ratio between the total simulation time spent in monomeric states following the dissociation of a spontaneously assembled dimer complex, and the total time the system spent in dimeric states*". The convergence of these estimates as a function of the simulation time is not provided (neither the error bars of these estimates) - at variance with the Lamprakis's and our work (see Fig. 1 below) - since their binding free energy estimates depend on the length of the simulation (defined by the user) as stated by the same authors who wrote at page 15 (*doi. 10.1371/journal.pcbi.1005169*) "*The binding free energies ΔG should be considered as lower bounds for the true binding free energies as the number of dissociation events decreases with increased simulation times*". We note that the calculation convergence is necessary to assess the quality of the results of the Böckmann's works because the binding free energy and the dimer states identified in their studies might change if the simulations are resumed and continued, which is not our case as shown by the free-energy convergence plots reported in Fig. 1 below.

The same authors confirm the dependence of their binding free energy estimates on user-defined parameters stating: "*For increasing dimerization threshold interaction energies (sum of Lennard-Jones and Coulomb interaction energies), less dissociations and thus increased binding free energy estimates are observed*." (Fig. S3 in *doi: 10.1371/journal.pcbi.1006062*). This should not happen. Binding free energy is a state function and its estimate should be independent from any user-defined parameter, while this is not their case inasmuch their binding free-energy estimates depend on the dimerization interaction energy threshold value set by the authors to 50 kJ/mol. In fact, from Fig. 8 of the same paper (*doi: 10.1371/journal.pcbi.1006062*), one can see that the definition of dimer states for CXCR5 is highly sensitive to the interaction energy threshold value (reported as "*dimerization criterium energy*" in the figure by the authors).

In another Böckmann's work where they used the same simulation protocol to investigate rhodopsin GPCR dimerization, the same authors admitted the limitation of their calculations writing *"The results clearly show that 500 times 1 μ s is not sufficient to get a converged view of association for this protein."* (doi. 10.1021/ct5010092).

The final result is that Böckmann and colleagues could observe several - but not all - dimer structures and could not provide their free-energy estimate. Proof is given by the fact that our lowest energy dimer structures were not found in their studies, neither among the less populated dimers reported in the Supplementary Information documents. Conversely, as can be seen from the FES shown in Fig. 2 below, we found as energetic metastable states all the CXCR4 dimer structures reported in the Böckmann study (doi. 10.1371/journal.pcbi.1005169).

Prompted by the Reviewer's comment, we have revised the main text at page 4 as follows to explain more clearly the importance of free-energy convergence and the calculation of relative and absolute binding free energy:

"In addition, the accuracy of the experimental technique - in our case the simulation model - should be also taken into account. Here, we employed the Martini coarse-grained force field⁵⁵⁻⁵⁷, which is known to be enthalpically driven with entropy loss due to the coarse-grained representation of the system⁵⁸⁻⁶⁰. This leads to over-stabilisation of the bound states. The employment of metadynamics in Martini coarse-grained simulations alleviates the issue, allowing the exploration of low probability regions⁶¹⁻⁶⁴. As a result, the diffusion of the proteins in the membrane is enhanced and the correct interaction between the proteins is preserved, including the structural and energetic information for high-energy states and unbound states. For this reason, CG-MetaD has been successfully used by us and other research groups to study protein/ protein interactions^{41,65,66}. The use of CG-MetaD ensures recrossing events between receptor dimer and monomer states, and a full exploration of the phase space (i.e., all the possible dimer states) leading to the convergence of free-energy calculation and a quantitatively characterised free energy surface. The latter is necessary to disclose the low energy - hence most probable - dimer states and the binding mechanism of these receptors, which is the aim of the present study. To this end, the relative free-energy difference between two or more dimer states allows the identification of the lower energy dimers, while the absolute binding free energy value has minor significance since it can be influenced by simulation conditions, including the accuracy of the force field. A detailed description of each dimeric structure is reported in the next section."

Figure 1. Evolution of the free-energy difference between bound and unbound state as a function of the simulation time. In the last 0.3 ms of simulation, highlighted in grey, the calculations converged leading to absolute binding free-energy estimates of -22.2 (+/- 0.3) kcal/mol for CCR5 homodimer, -21.1 (+/- 0.2) kcal/mol for CXCR4 homodimer, and -24.5 (+/- 0.2) kcal/mol for CCR5-CXCR4 heterodimer.

Figure 2. Representation of CXCR4 dimer structures adapted from the study of Pluhackova et al. (*doi: 10.1371/journal.pcbi.1005169*) onto the FES computed with CG-MetaD. The positioning of the dimers in the FES might be inaccurate since the dimers were selected based on their images reported in Fig. 2 of Pluhackova et al..

2) Here, a reference to Martini 3 would be appropriate, in particular, since Martini 3 was shown to yield results in favorable agreement with experiments (Souza et al. 2021. *Nat. Methods*. 18:382–388). Martini 3 was seemingly not used here as suggested by the following sentence in the Results and Discussion section, lines 146,147: 'Here, we employed the Martini coarse-grained force field (55–57),..'

We thank the Reviewer for the suggestion. We collaborate with the developers of the Martini force field for other projects and we recently published together in *Nature Communications* a work on ligand/protein binding with CG simulations (doi: 10.1038/s41467-020-17437-5). At the time of this study, Martini 3 was under development and still during the review process the cholesterol parameters were not available. These parameters were released only last month, albeit the related paper is still preprinted (<https://chemrxiv.org/engage/chemrxiv/article-details/64677ce5fb40f6b3eed8bb26>). Thus, we have used Martini 3 to run the additional simulations suggested by the Reviewer at the next point. We anticipate that no significant change is found by using Martini 2 or 3. In addition, Lamprakos et al. (doi: 10.1021/acs.jctc.0c00507) recently performed a comparison study between these two versions of the Martini force field and interestingly they wrote “*Martini 3 reparameterization was performed with an eye to adjust overestimated dimerization barriers using Martini 2.2P. However, we observe that the estimated energies now underestimate the experimentally calculated free energy of dimerization by a considerable amount in all of the cases that we studied with the exception of the EphA1 dimer.*” Based also on our experience in ongoing studies where we are using Martini 3, CG models continue being enthalpically driven and the combination of CG with MetaD remains a most successful strategy to enhance the phase space exploration.

The use of Martini 3 for the additional simulations is now reported in the revised manuscript at page 14:

“Furthermore, additional unbiased CG-MD and CG-MetaD calculations on CCR5-CCR5 binding in the plasma membrane model were performed using the latest version of the Martini force field (Martini 3)⁸⁰, which was released during the review process of the present article. As can be seen from Fig. S13, the exploration of the phase space remains remarkably superior with CG-MetaD if compared with that of unbiased CG simulations, including the FES region with the low energy minima.”

3) *It is great to see that the authors added in the revision simulations employing an improved biomembrane model. However, instead of investigating the chemokine receptor dimerization in this biomembrane model (the topic of this study), the authors instead studied the 'stability' of previously identified dimers within this model membrane employing standard Martini simulations. It is well known that Martini transmembrane dimers tend to be glued together, that observing dissociation is highly unlikely even if the dimer would not spontaneously form within this environment. Interestingly, the authors show themselves in Figure S10 that this is not the way to go. It was already shown before that GPCR dimerization may be substantially modified in different lipid environments (probably not yet for chemokine receptors).*

We agree with the Reviewer that Martini force field is enthalpically driven and might over-stabilise the bound states. That is the reason why we have developed and use CG-MetaD as explained at previous Point 1. However, we note that using Martini CG force field and unbiased MD calculations, unstable dimer structures can leave the starting pose and move to alternative, more stable dimer conformations. We demonstrated this is the case of the simulations starting from the X-ray dimer structures shown in Fig. S10 and even other authors, including Böckmann, observed that receptor dissociation might occur within few ts of standard CG simulations (please see Fig. 7 of [doi: 10.1371/journal.pcbi.1005169.g007](https://doi.org/10.1371/journal.pcbi.1005169.g007)).

Therefore, 100 ts stability of our dimers structures in the plasma membrane model with 30% of cholesterol is in our opinion a convincing evidence. This result is relevant further considering that the presence of diverse phospholipids and 30% cholesterol concentration might destabilise receptor dimers by intercalating lipids between protomers, as also noted in the Böckmann's studies. In our study, this occurs in two cases, CCR5 dimer A and CCR5-CXCR4 dimer B, where DOPE and cholesterol lipids find places at the protomer binding interface, however maintaining the dimer binding mode.

Nevertheless, we have acknowledged the comment of the Reviewer performing additional calculations. In particular, since atomistic force field is more accurate and not limited by over-stabilisation of bound states with respect to Martini CG force field, we performed in the plasma membrane model 12 ts of classical all-atom MD calculations on the six CCR5 and CXCR4 dimers identified in our study. The results confirm their energetic and structural stability with a low RMSD value of 0.10 nm computed for the receptors backbone atoms (see Fig. 3 below).

Furthermore, we have performed additional unbiased CG and biased CG-MetaD calculations on CCR5 binding in the plasma membrane model and using the recently released Martini force field version 3. As can be seen from Fig. 4 below, after only 30 ts of simulations, the phase space sampling of CG-MetaD is remarkably superior to that of unbiased CG, as expected, and interestingly the FES regions of dimer A and B have already started being explored. We note that this FES is rough since the free-energy calculation is far from convergence that might take ms (months of simulations), therefore their completion might be argument of a future work.

The results of these additional simulations are reported at page 14 of the revised manuscript and below:

“The structural stability of the six dimers identified for CCR5 and CXCR4 has been further assessed in the plasma membrane model by means of 12 μ s atomistic MD calculations. During these simulations, the binding mode between receptors in all the dimer structures is stable with a low average RMSD of 0.10 nm computed for the receptors backbone atoms (see Fig. S4). Furthermore, additional unbiased CG-MD and CG-MetaD calculations on CCR5-CCR5 binding in the plasma membrane model were performed using the latest version of the Martini force field (Martini 3)⁸⁰, which was released during the review process of the present article. As can be seen from Fig. S13, after only 30 μ s of simulations the phase space exploration by means of CG-MetaD remains remarkably superior if compared with that of unbiased CG simulations, including the FES region of the low energy minima.”

Figure 3. Plots of the RMSD computed for the C α atoms of CCR5 and CXCR4 during the atomistic MD calculations in plasma membrane on the CCR5 and CXCR4 homo- and heterodimers identified by CG-MetaD calculations.

Figure 4. Top) Phase space exploration represented as light grey spots during 30 I.Ls unbiased CG-MD (left) and CG-MetaD (right) calculations in CCR5-CCR5 system. Bottom) The converged dimerization FES reported in the manuscript (left) and the FES obtained after 30 I.Ls of CG-MetaD in plasma membrane model using the recently released Martini 3 force field.

4) *Diving into the previously published work* - Reviewer refers to the Böckmann's work with doi: 10.1371/journal.pcbi.1005169-, both symmetric and asymmetric CXCR4 dimers were described previously - similar to this work. The TM1/TM5-7 probably corresponds to the TM1-TM6-TM7 binding interface here. TM6 participation in the interface is thus not a novel observation like suggested by the results description (page 8). Regarding the cholesterol- stabilized interface: A careful comparison shows that the TM3-TM4 was dominant only for 30% cholesterol in Pluhackova et al. At 10% cholesterol, this interface rarely occurred suggesting a lower binding affinity, similar to what is seen here.

We respectfully disagree. All the lowest energy dimer structures identified in our study were not found in the Böckmann's studies (doi: 10.1371/journal.pcbi.1005169 and doi: 10.1371/journal.pcbi.1006062), neither among the less populated dimers reported in Supplementary Information (please see Fig. S4 of doi: 10.1371/journal.pcbi.1006062).

Regarding the TM1/TM5-7 dimer, the same authors refer to this structure as “*asymmetric TM1/TM5-7 dimer*”. So, how can this state be similar to the symmetric TM1-TM6-TM7 dimer? However, to fulfill the Reviewer’s comment, we have prepared the following Fig. 5 that clearly shows the diverse binding interface between our symmetric TM1-TM6-TM7 dimer and the asymmetric TM1/TM5-7 dimer from Pluhackova et al. (*doi: 10.1371/journal.pcbi.1005169*), with a RMSD value of 1.07 nm between the two structures. No further explanation is necessary.

Finally, we remark that we have not reported that TM6 participation in CXCR4 dimerization is a novel observation. On the contrary, at page 8 of the original manuscript we wrote “*Furthermore, residues of TM6 engage favorable interactions in both structures A and B, supporting previous evidence showing that TM6 participates in the stabilization of CXCR4 homodimer^{27,36}.*”.

Figure 5. Comparison between the TM1-TM6-TM7 dimer structure (left) identified in our study and the TM1/TM5-7 dimer (right) reconstructed from Fig. 3 of Pluhackova et al. (*doi: 10.1371/journal.pcbi.1005169*). The dimers are shown as top-view cartoon (top) and cylinders (middle), and side-view cartoon (bottom). In the top-view cartoon representation, the loops were omitted for clarity. The helices are coloured according to the legend reported at the bottom of the figure.

5) In a second work, the same authors investigated CCR5 and CCR5-CXCR4 dimerization.⁵² Here, the most populated dimer interfaces are asymmetric where CCR5 forms dimers through TM1-H8/TM4-TM5 and TM1-H8/TM5-TM6-TM7, whereas CCR5 and CXCR4 interact through TM1 and TM4-TM5-TM6-TM7. At variance with these results, we show that the lowest energy CCR5 and CCR5-CXCR4 dimers have symmetric binding interface composed of TM4-TM5, however low energy asymmetric dimers are also possible through TM1-TM2-H8/TM4-TM5 and TM1-TM2-H8/TM6-TM7 for CCR5 and CCR5-CXCR4 dimers, respectively. The different results might be attributed to the diverse simulation technique employed.

Reviewer wrote: “The authors compare with a dimerization study performed in either 100%POPC or in POPC/30% cholesterol, i.e. a different lipid environment than the one studied here.”

We have acknowledged the Reviewer’s comment revising the sentence at page 18 as follows:

“The different results might be attributed to *the different cholesterol concentration of the membrane and* the diverse simulation technique employed.”

6) For simplified comparison and understanding of the energy landscapes it would be great if the authors could explain in a sketch how the one-dimensional torsion angle can be used to characterize the two-dimensional orientation space for two proteins forming a dimer.

The representation of the torsion CV used in CG-MetaD calculations was already reported in the original manuscript in Fig. S1. For the Reviewer convenience, we report also this figure below (Fig. 6). In addition, to acknowledge the Reviewer’s comment we have prepared two additional figures:

- Fig. 7 showing the CXCR4 dimerization FES with minimum states having different torsion CV values;
- Fig. 8 showing the dimer structures representing the most populated clusters extracted from the diverse minima labelled in Fig. 7, and a table reporting the RMSD values between the diverse structures computed for the C α atoms of the transmembrane helices.

From this analysis, in particular from Fig. 8, it can be seen that the torsion CV well discriminates the different dimer states assumed by the CXCR4 receptor.

Figure 6. The center of mass of the CG backbone beads (BB) of the residues indicated in the figure was used to define the torsion CV. The C, D, E, and F points were used to define the torsion angle describing the relative orientation between the two protomers.

Figure 7. The FES of the CXCR4 homodimerization. The black solid line represents one identified lowest energy path from the monomer (unbound) to the dimer (bound) state. The white labels highlight the position of the energy minima.

Figure 8. Comparison between the dimer states representing the most populated clusters extracted from the different minima. Upper) Top view of dimers A, B, c1, c2, δ and ε, represented as colour-coded cylinders. Lower) Table reporting the RMSD values between the dimers computed on the Ca atoms of the TMs.

7) In fact, both in ref 51 and 52 the authors performed standard coarse-grained molecular dynamics simulations, which allow only a partial exploration of the dimerization free energy surface as also shown by our simulations reported in the Supplementary Materials (see Supplementary Figure S10).

Reviewer wrote: This is not shown in Figure S10. Figure S10 only compares the limitation of starting from a preformed dimer (from crystal structure). It basically shows that the simulations using a complex lipid composition added in the revision are of little value if one is interested in dimerization. References 51 and 52 instead employ hundreds of simulations starting from different structures with well separated chemokine monomers.

Following the Reviewer's request we have removed the reference to Fig. S10 and revised the sentence as reported below. In particular, we point out that using standard CG calculations, despite a huge number of simulations, one might not achieve an exhaustive exploration of the phase space. A full sampling of the phase space should be demonstrated with the convergence of the results. This implies that the identified dimer states and the binding free energy estimates do not change as a function of the simulation time - in other words they do not change if the simulations are resumed and continued, then the calculation is converged. Sadly, this is not the case of ref. 51 and 52. A thorough exploration of the phase space can instead be obtained employing CG-MetaD. Please see also reply to point 1.

The revised sentence is as follows:

“In fact, both in ref. 51 and 52 the authors performed hundreds of standard coarse-grained molecular dynamics simulations, without however reaching a full exploration of the dimerization free energy landscape. Differently, here we have combined metadynamics with coarse-grained molecular dynamics. This allows enhancing the sampling of the phase space and achieving a converged dimerization free-energy landscape.”

The sentence is now much clearer, thank you!

8) Differently, here we have combined metadynamics with coarse-grained molecular dynamics. This allows enhancing the sampling of the phase space and achieving a converged dimerization free-energy landscape. In so doing, each possible dimer state is visited and energetically evaluated - including those proposed in ref. 51 and 52 - providing a thorough description of receptor dimerization mechanism. Proof of that is further given by the fact that the X-ray dimer structures of CCR5 and CXCR4 are also identified in our study (see Fig. 5), while they are not found by standard CG- MD simulations reported in refs 51 and 52.”

Reviewer wrote: Again, this assessment of 51,52 is not correct. Rmsd to crystal dimer structures of 3.9Å and 4.1Å were reported in Pluhackova et al.

We note that the rmsd values of 3.9 and 4.1 Å from the experimental CXCR4 dimers reported by Pluhackova et al. ([doi. 10.1371/journal.pcbi.1005169.g002](https://doi.org/10.1371/journal.pcbi.1005169.g002)) are rather high to confidentially conclude that two protein structures are similar. In addition, the X-ray structures of CCR5 were not found in their simulations as acknowledged by the same authors who wrote at page 9 of their manuscript ([doi. 10.1371/journal.pcbi.1006062](https://doi.org/10.1371/journal.pcbi.1006062)) *“The CCR5 TM1,H8/TM4,5 dimer configuration deviates by 6.7 Å (backbone RMSD of transmembrane region) from the dimer configuration observed in the CCR5 crystal structure.”*

Nevertheless, we have acknowledged the Reviewer's comment removing the reference to the CXCR4 crystal structure and revising the text as follows:

“Differently, here we have combined metadynamics with coarse-grained molecular dynamics. This allows enhancing the sampling of the phase space and achieving a converged dimerization free-energy landscape. In doing so, each possible dimer state is visited and energetically evaluated - including those proposed in ref. 51 and 52 - providing a thorough description of receptor dimerization mechanism. Proof of that is further given by the fact that the X-ray dimer structure of CCR5 (PDB ID 4MBS) is identified in our study (see Fig. 5), while it is not found by standard CG-MD simulations reported in ref. 52.”

Reviewer 4

1) In addition, the authors state that the receptors studied “prefer forming dimers instead of being in monomeric states” (line 141). And they say that this is in agreement with the results of the Mellado group. This statement is highly misleading because the Mellado group’s results on GPCR dimerization have been heavily debated. In contrast, Lohse’s group provides compelling evidence that CXCR4 receptors form mostly monomers at the cell surface when expressed under physiological (low amount of receptors) conditions (Isbilir et al (2020 PNAS; PMID. 33148803). Unfortunately, this study and reference are not discussed here.

We thank the Reviewer for raising this important point. Our results are in agreement with those from the Lohse’s study (Isbilir et al. doi. 10.1073/pnas.2013319117) that was indeed cited and discussed in three different sections of the original manuscript, at page 2 (Introduction), page 8 (Results), and page 18 (Conclusions). For the Reviewer convenience, we report below the original text:

“Particularly for CXCR4, recent works showed a correlation between the expression levels of the receptor with the dimers formation^{36,37}, whereas specific CXCR4 ligands - AMD3100, IT1t - can modulate the dimer vs. monomer equilibrium³⁶⁻³⁹. ”

“Furthermore, residues of TM6 engage favorable interactions in both structures A and B, supporting previous evidence showing that TM6 participates in the stabilization of CXCR4 homodimer^{27,36}. In particular, mutations on TM6 like V242D^{6,38} and L246P^{6,42}, both at the binding interface in dimer structure B (Fig. 2B), were found by Isbilir et al. to inhibit the formation of homodimers and reduce the receptor basal activity³⁶.” (red text added in the current revision).

“The same applies to CXCR4, in which TM6-TM7 forms the binding site in both the symmetric and asymmetric dimeric structures A and B, whereas TM4-TM5 is involved only in the dimeric structure B (Fig. 2B). This observation assumes even more relevance in light of the works of Martínez-Muñoz et al.²⁷ and Isbilir et al.³⁶ showing that the CXCR4 triple mutant K239E^{6,35}-

V242A^{6,38}-L246A^{6,42} on TM6 does not form receptor oligomers, while it is still able to form dimers.
”

It is interesting to note that Isbilir et al. found CXCR4 mainly in the monomeric form (86.7% vs. 11.6%) only “at the low expression levels [**below 0.3 receptors per μm^2**] that are required for single-molecule experiments“, while “the highly dimeric nature of CXCR4 at densities $>50,000$ receptors per cell (**>70 receptors per μm^2** membrane) might as well represent the scenario of CXCR4 organization in oncogenic cells.”, as stated by the authors. They further reported that high receptor densities might also be found under physiological condition considering that “CXCR4 expression in blood cells ranges from a few thousand to $>100,000$ receptors per cell [which corresponds to **<4 to >300 receptors per μm^2**]”.

For simulation purpose, we have 2 receptors in a membrane model of 400 nm^2 , which corresponds to **$\sim 5,000$ receptors per μm^2** . As another example, in the Bockmann’s study the receptor density is **$15,000$ per μm^2** (receptor density value of 0.015 nm^{-2} taken from Table 1 of doi. 10.1371/journal.pcbi.1006062). Such a high receptor concentration favours the formation of dimers as also reported in the Lohse’s study. Furthermore, chemokine receptors are expressed in diverse tissues, and factors like receptor density (e.g., receptor expression in diverse tissues), membrane composition (lipid types and presence of diverse proteins) and specific cell condition (physiological vs. pathological, cellular stress) might significantly influence receptors interaction as also reported by Lohse and colleagues.

Taking such aspects into account, we have revised the text at page 4 as follows:

“We note that our simulation model embeds 2 receptors in a membrane of 400 nm^2 , resulting in an approximate density of $\sim 5,000$ receptors per μm^2 . Interestingly, Isbilir et al.³⁶ found that CXCR4 shows a highly dimeric nature already at receptor densities higher than 70 receptors per μm^2 , while the monomeric state is favoured at receptor densities below 0.3 receptors per μm^2 . Consistent with these findings, our results confirm that at high receptor density these GPCRs prefer forming dimers rather than remaining in the monomeric state. A similar trend was also found for the same receptors in T cells experiments where dimer and oligomer forms were largely more present than the monomer state^{26,27}. However, we note that comparing data from different studies is not trivial since factors like receptor density (e.g., receptor expression in diverse tissues) and the membrane composition (lipid types and presence of diverse proteins) might influence the receptors binding interaction.”

We thank again the Reviewer for raising this important point.

2) Line 119-120: when the simulations started, the two protomers were placed about 7 nm apart. Could you show a picture of the starting situation in a figure, to better evaluate this distance in relation to the size the protomers?

We have acknowledged the Reviewer's request preparing the following picture that is included in the revised Supplementary Materials as Fig. S14.

Figure 9. Representation of chemokine GPCRs in the unbound state at the beginning of the CG-MetaD simulations.

3) In most of the figures, plot axis legends are too small to be read.

We have increased the axis label size making all the figures readable.

4) Paragraph on “Dimerization mechanism” (line 352-404) is written in too much detail. As a result, it is difficult to get a clear picture.

Following the Reviewer's suggestion, we have revised the “Dimerization mechanism” section reporting only the major findings in the main text, while further details can be found in Supplementary Materials.

5) The end of the introduction is very long (line 81-104).

Prompted by the Reviewer's comment, we have revised and shortened the end of the Introduction.

6) Line 114: delete “in Nature Methods”.

Done.

7)The G protein should be named “Galphabetagamma” and not “Galphabetadelta”.

Done.

8)The expression “virtual microscope” (line 69) is not adapted as proposed in my review of the original manuscript.

Prompted by the Reviewer’s comment, the sentence:

“In detail, we generated a virtual microscope in which the receptors are immersed in a POPC membrane model with 10% of cholesterol.”

has been replaced by:

“In detail, we generated a model in which the receptors are immersed in a POPC phospholipid bilayer with 10% of cholesterol.”

We thank again all the Reviewers for their work and useful suggestions. We believe that the revised manuscript is now ready for publication in *Nature Communications*.

Looking forward to hearing from you.

Sincerely,

Vittorio Limongelli

REVIEWERS' COMMENTS

Reviewer #3 (Remarks to the Author):

The authors substantially improved the clarity of the manuscript, they did a great job! Thanks!

Reviewer #4 (Remarks to the Author):

The current revised version of the manuscript has addressed my concerns.